# Lateralized hippocampal oscillations underlie distinct aspects of human spatial memory and navigation

Jonathan Miller[1], Andrew J. Watrous[1], Melina Tsitsiklis[2], Sang Ah Lee [3], Sameer A. Sheth[4], Catherine A. Schevon[5], Elliot H. Smith[6], Michael R. Sperling[7], Ashwini Sharan[8], Ali Akbar Asadi-Pooya[7,9], Gregory A. Worrell[10], Stephen Meisenhelter[11], Cory S. Inman[12], Kathryn A. Davis[13], Bradley Lega[14], Paul A. Wanda[15], Sandhitsu R. Das[13], Joel M. Stein[16], Richard Gorniak[17] & Joshua Jacobs[1]

The hippocampus plays a vital role in various aspects of cognition including both memory and spatial navigation. To understand electrophysiologically how the hippocampus supports these processes, we recorded intracranial electroencephalographic activity from 46 neurosurgical patients as they performed a spatial memory task. We measure signals from multiple brain regions, including both left and right hippocampi, and we use spectral analysis to identify oscillatory patterns related to memory encoding and navigation. We show that in the left but not right hippocampus, the amplitude of oscillations in the 1–3-Hz "low theta" band increases when viewing subsequently remembered object–location pairs. In contrast, in the right but not left hippocampus, low-theta activity increases during periods of navigation. The frequencies of these hippocampal signals are slower than task-related signals in the neocortex. These results suggest that the human brain includes multiple lateralized oscillatory networks that support different aspects of cognition.

[1] Department of Biomedical Engineering, Columbia University, 351 Engineering Terrace, Mail Code 8904, 1210 Amsterdam Avenue, New York, NY 10027, USA. [2] Doctoral Program in Neurobiology and Behavior, Columbia University, New York 10027 NY, USA. [3] Department of Bio and Brain Engineering, Korea Advanced Institute of Science and Technology, Daejeon 34141, Korea. [4] Department of Neurosurgery, Baylor College of Medicine, Houston 77030 TX, USA. [5] Department of Neurology, Columbia University Medical Center, New York 10032 NY, USA. [6] Department of Neurological Surgery, Columbia University Medical Center, New York 10032 NY, USA. [7] Department of Neurology, Thomas Jefferson University, Philadelphia 19107 PA, USA. [8] Department of Neurosurgery, Thomas Jefferson University, Philadelphia 19107 PA, USA. [9] Shiraz Neurosciences Research Center, Shiraz University of Medical Sciences, Shiraz 71348, Iran. [10] Department of Neurology, Mayo Clinic, Rochester 55905 MN, USA. [11] Department of Neurology, Geisel School of Medicine at Dartmouth, Lebanon 03756 NH, USA. [12] Emory University School of Medicine, Atlanta 30322 GA, USA. [13] Department of Neurology, Hospital of the University of Pennsylvania, Philadelphia 19104 PA, USA. [14] University of Texas–Southwestern, Dallas 75390 TX, USA. [15] Department of Psychology, University of Pennsylvania, Philadelphia 19104 PA, USA. [16] Department of Radiology, Hospital of the University of Pennsylvania, Philadelphia 19104 PA, USA. [17] Department of Radiology, Thomas Jefferson University, Philadelphia 19107 PA, USA. Correspondence and requests for materials should be addressed to J.J. (email: joshua.jacobs@columbia.edu)

Episodic memory, the ability to remember life's daily episodes, has garnered intense research interest over recent decades. Foundational research has shown that although episodic memory involves widespread brain regions[1], the hippocampus in particular is vital[2]. Thus, a key issue in understanding the neural basis of episodic memory is characterizing how the hippocampus coordinates the brain-wide networks where memories are eventually stored. One phenomenon that may underlie memory formation is the theta oscillation. In rodents, the hippocampus exhibits theta oscillations at 4–8 Hz whenever this structure is active[3]. The theta oscillation in rodents is specifically hypothesized to be involved in memory because its amplitude correlates with memory encoding[4], its timing modulates synaptic plasticity[5], and it is known to synchronize activity across brain-wide neural networks[6,7]. There is also evidence that the role of hippocampal theta extends beyond memory into other behaviors[8]. Theta oscillations correlate with movement during spatial navigation in rodents[9,10] and humans[11,12], and theta underlies the representation of location by entorhinal grid cells[13]. Theta is also considered to be important for representing location during navigation[14,15].

How could a single electrophysiological pattern like the theta oscillation appear to support such a diverse range of neurobehavioral processes, including memory and path integration? Recent work has raised the possibility that there are, in fact, multiple theta oscillations in a given individual[7,16]. In a separate line of inquiry, neuroimaging research suggested that the two hippocampi indeed have separate functional roles[17,18] and, in particular, the notion has emerged that activity in the left and right hemispheres separately correlate with verbal and spatial processing, respectively[19,20]. Drawing on this diverse body of work, we considered that human hippocampal theta oscillations actually consist of lateralized signals across left and right hemispheres that each support different behavioral and cognitive processes. This idea that theta has multiple subcomponents differs from the view that hippocampal theta is a unitary signal that is similar across hemispheres, as suggested by rodent studies[21–23].

We investigate these issues by examining direct brain recordings from neurosurgical patients who performed a hybrid navigation and memory experiment called Treasure Hunt (TH). TH draws inspiration from both the human verbal memory and rodent spatial navigation domains and asks subjects to memorize multiple object locations while virtually navigating an open arena. As such, TH can be thought of as a spatial paired-associate learning task in which participants memorize object–location pairs—this type of task is known to be dependent on the hippocampus[24]. TH's design includes separate time intervals for memory encoding and navigation, allowing us to better distinguish the neural correlates of these processes. To our knowledge, our study is the largest current investigation of the direct electrophysiological correlates of human spatial memory—including 100 memory-encoding events per session—which provides more data per session compared to earlier paradigms[25]. By analyzing the relation between task behavior and oscillatory power at various frequencies, we sought to elucidate the electrophysiological basis of human spatial navigation and memory, including understanding functional differences between the hippocampus and neocortex, and between the left and right hemispheres. Here,

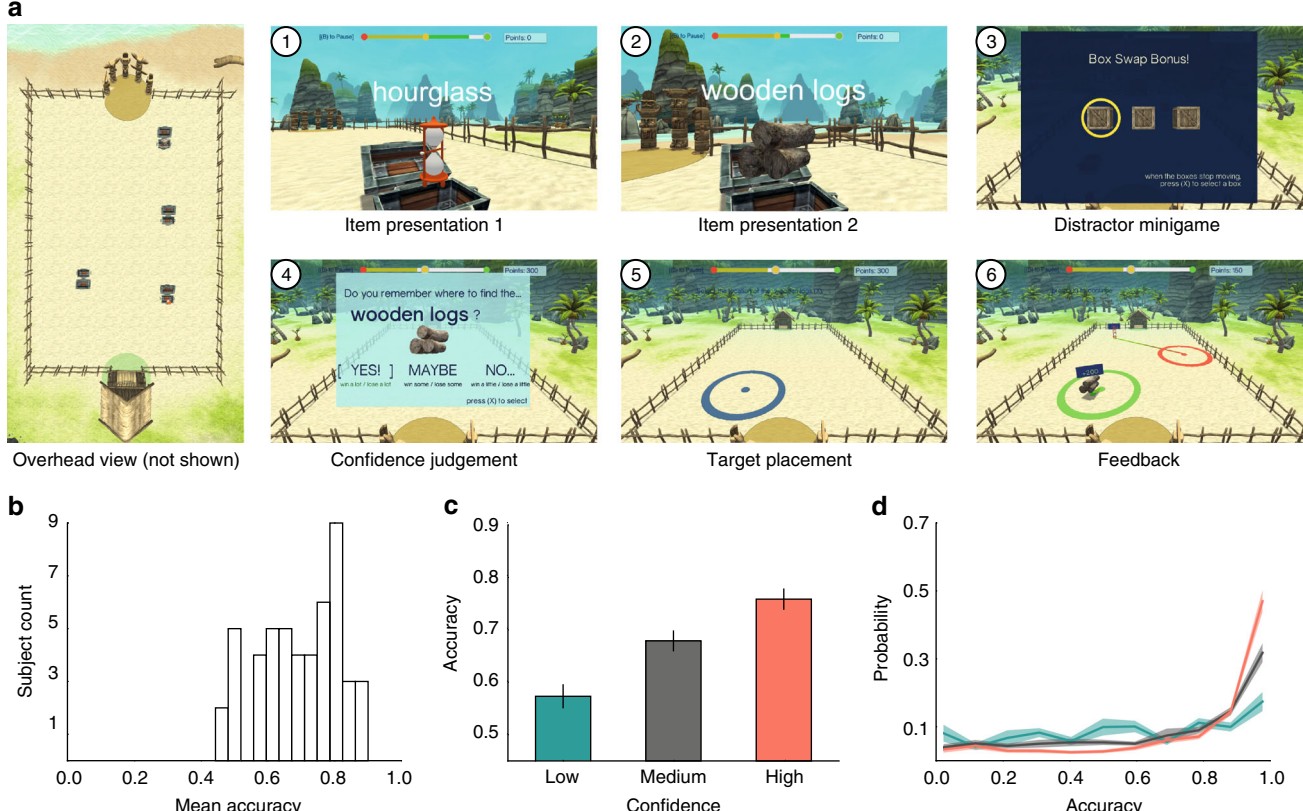

**Fig. 1** Task methods and behavior. **a** Screen shots of an example trial in the Treasure Hunt task. The top left image shows an overhead view of the environment with chest positions visible. This view is never shown to subjects, and only one chest is visible at a time during the actual trial. Panels 1–6: representative epochs of a single trial, showing 1) first item presentation, 2) second item presentation, 3) the distractor period before the retrieval period, 4) the confidence judgement, 5) responding by indicating an item's remembered location, and 6) feedback. Treasure Hunt was created using the Unity 3D graphics engine. **b** Histogram of mean subject accuracy ($N = 50$). **c** Mean accuracy as a function of subjective confidence. Error bars are ±1 SEM. **d** Probability of a given response as a function of accuracy, calculated independently for each confidence level. Error bars are ±1 SEM

we show that increased 1–3-Hz power correlates with both successful memory encoding and navigation in the hippocampus and the lateral temporal lobe. This slow-theta band, although slower than the range where such signals are present in rodents, has previously been shown to show task-related activity in the human hippocampus[26]. Notably, in the hippocampus the memory effect is lateralized to the left hemisphere, whereas navigation-related activity is prominent in the right hemisphere. Our findings provide electrophysiological evidence for the lateralization of human hippocampal function in memory. Moreover, although we observe memory-related activity in the neocortex, the frequency of this signal differed compared to the hippocampus. Together, our findings indicate that multiple oscillatory networks support different aspects of cognition throughout the brain.

## Results

**Direct human brain recordings in a spatial memory task.** To examine the neural basis of human spatial memory and navigation, we asked epilepsy patients with surgically implanted electrodes to perform our TH spatial memory task. In each trial of the task, patients learned the locations of several objects in a large rectangular arena on a virtual beach (Fig. 1a). During learning, patients drove to a series of treasure chests, each of which was positioned at a different random location. When the patient reached each chest, it opened, revealing an object whose location they needed to remember. The object remained visible for 1500 ms and then disappeared. The patient then drove to the next chest. After a series of these learning events, the retrieval phase began. During retrieval, patients were shown the name and image of each object and asked to respond by indicating the location where that object had been encountered. Our analyses characterized neural signals during the learning of object–location pairs and

during traversals to treasure chests to reveal the neural basis of spatial memory encoding and navigation.

**Analysis of behavioral performance in the task.** We assessed task performance by measuring the patient's accuracy for each studied item (Fig. 1b–d). We computed the distance between the response location of each item and its actual position, and we normalized the distance to account for the distribution of possible response locations (see Methods and Supplementary Figure 1). As a result of this normalization, our accuracy measure ranges between 0 (worst possible response) and 1 (best possible). Figure 1b shows a histogram of each subject's mean normalized accuracy ($N = 50$, median $= 0.70$, mean $= 0.69$, SEM $= 0.017$). We also asked subjects to indicate their subjective confidence for each remembered item using a three-point scale. Figure 1c, d show accuracy as a function of subjective confidence. Accuracy significantly improved with increasing confidence (one-way ANOVA, $F(2, 135) = 18.34$, $p < 10^{-7}$).

**Identifying neural signals related to memory encoding.** We analyzed direct neural recordings from 5807 electrodes across 46 subjects, including both surface and depth contacts, to characterize neural signatures associated with successful encoding of object locations. The recordings sampled a range of brain areas (Fig. 2), including left and right hippocampi. Electrode coverage included 79 left hippocampal electrodes from 26 subjects and 55 right hippocampal electrodes from 21 subjects. 10 subjects had bilateral hippocampal implants and 27 had electrodes in only one hemisphere.

We identified the segments of the recordings corresponding to when the patients studied an item at each individual treasure chest. To distinguish brain signals that differentiated successful

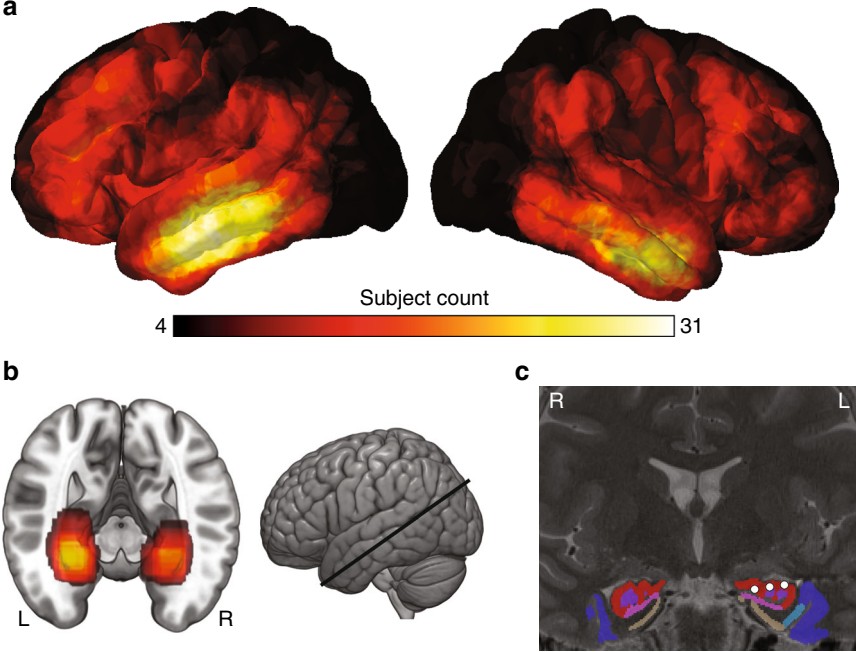

**Fig. 2** Electrode characteristics. **a** Colored brain surfaces showing the number of subjects contributing electrodes to each neocortical location. Electrodes contribute to a location if they are within 12.5 mm of a given point on the brain surface. Black coloring indicates coverage from four or less subjects. **b** A cross section along the longitudinal axis of the hippocampus (slice location shown to the right) indicating the number of subjects contributing electrodes to each location. Electrodes contribute to a location if they are within 3 mm of a given voxel. **c** Coronal MRI image showing electrode positions from one example patient with depth electrodes. Individual medial temporal lobe (MTL) regions are colored using our automated segmentation procedure. Three electrodes in region CA1 (red) are shown in white. CA1: red, subiculum: pink, dentate gyrus: purple, entorhinal cortex: tan, BA 35: light blue, BA 36: dark blue

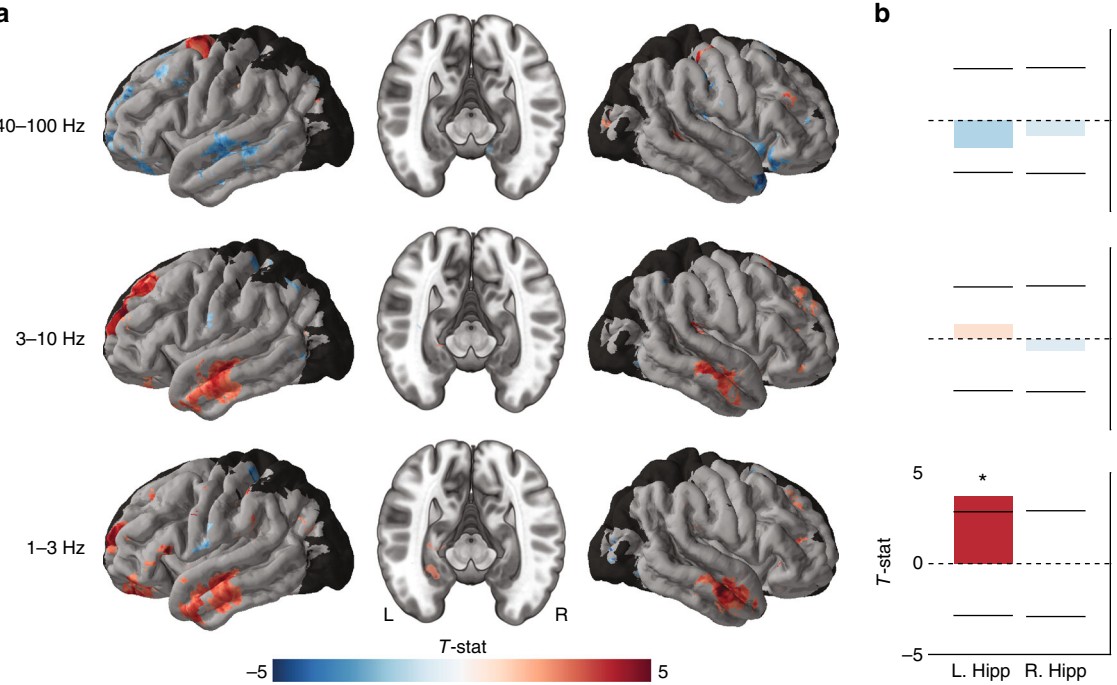

**Fig. 3** Group level memory-related changes in spectral power in three frequency bands. Red indicates greater power for viewing an item that was subsequently recalled compared to an item that was not recalled, blue indicates greater power for not recalled items. Color corresponds to the *t*-statistic from comparing the distribution of subject-level effects to zero. **a** Cortical surface plots and hippocampal cross sections of significant changes in memory-related power during item encoding, thresholded using a permutation procedure (see Methods). Non-significant areas are in gray. Neocortical regions with less than five subjects were not included and are rendered in black. Cross-sectional plots show only data from electrodes localized to the hippocampus. **b** *T*-statistics averaged across the whole left (*N* = 26) and right (*N* = 21) hippocampi. Horizontal bars represent the critical *t* value for significance, determined by the degrees of freedom of that ROI and a Bonferroni correction

from unsuccessful memory encoding, we labeled each encoding event according to whether the viewed item was successfully remembered or if it was forgotten (see Methods). We measured the power of oscillatory neural activity during each encoding event in each of three bands—1–3 Hz (low theta), 3–10 Hz (theta), and 40–100 Hz (high-frequency activity, HFA)—and averaged across the 0–1500-ms period relative to item appearance. We refer to this 1–3 Hz band as "low theta" as opposed to the different nomenclature "delta" because the human hippocampus exhibits navigation-related[27,28] and memory-related oscillations[29,30] in this band, suggesting that this signal is analogous, at least in part, to the ~8-Hz theta oscillations seen in rodents[26].

To identify neural signatures of successful memory encoding, we used *t* tests to compare the power distributions for memory encoding between the remembered and forgotten items, at each band or frequency. As in studies of verbal memory[29,31], many electrodes significantly varied in power during encoding according to whether a viewed item would be subsequently remembered. Overall, the general trend across the whole dataset was that power in the low-theta and theta bands was elevated for recalled compared to forgotten items. In the neocortex, this theta effect was most prominent for electrodes in the anterior lateral temporal lobe, with weaker effects in the HFA band or in other surface regions (Fig. 3a). We next turned our attention to depth electrodes, given our interest in the role of medial temporal lobe (MTL) structures in spatial memory. In the left hippocampus, we found a 1–3 Hz power increase for recalled items compared to non-recalled items (one-sample *t*-test, *t*(25) = 3.72, *p* < 0.01). There were no significant memory effects in the right hippocampus at any frequency band (Fig. 3b).

To illustrate these memory-related power changes more precisely, Fig. 4a depicts the relation between oscillatory power and memory encoding as a function of frequency and time for the left and right hippocampus and lateral temporal lobe. The left hippocampus shows increased 1–3-Hz power for items that were successfully remembered beginning ~500-ms before item presentation and continuing through the presentation interval. In contrast, the right hippocampus does not show such clusters of activity. These group-level effects are also visible in individual electrodes and at the single-trial level (e.g., Fig. 4b, see Supplementary Figure 2). These plots demonstrate a fundamental difference in the spectral characteristics of the memory-related oscillatory activity between the hippocampus and neocortex. In the left lateral temporal lobe, the low-frequency oscillations related to memory had a broad frequency bandwidth up to ~10 Hz. This signal thus encompassed a substantially wider range than the hippocampal signals, which were limited to frequencies below 4 Hz.

We considered the possibility that apparent memory-related neural signals were actually related to variations in difficulty for remembering objects in particular locations in the environment. Behaviorally, subjects were more accurate in locating items encountered in the near half of the field relative to the testing location compared to the far half (0.71 vs. 0.67; paired-sample *t*-test, *t*(49) = 4.9, *p* < 0.001). However, this behavioral difference did not manifest as differences in neural activity, as the magnitude of the left-hippocampal low-theta memory effect was similar for both near and far items (paired-sample *t*-test, *p* > 0.1). Similarly, subjects exhibited more accurate memory performance for items studied near the boundaries of the environment compared to items studied closer to the center (0.71 vs. 0.66,

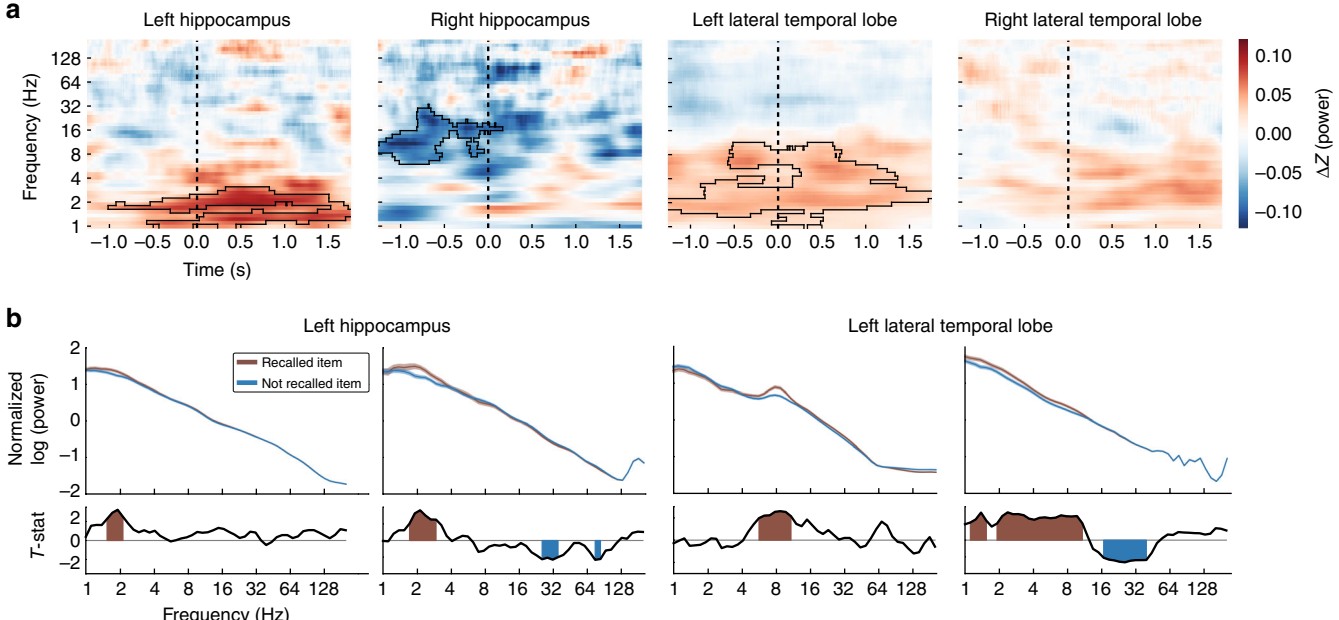

**Fig. 4** Illustration of subsequent memory effects in hippocampus and temporal lobe. **a** Group level time–frequency spectrograms, showing mean difference in normalized power between recalled and not-recalled stimuli. Changes in power are shown at each of 50 log-spaced frequencies between 1 and 200 Hz and for 69 100-ms time bins spanning −1.5 to 2 s relative to item onset. Black outlines indicate significant clusters of changes in power (nonparametric clustering procedure[33] $p < 0.05$). The dashed vertical line indicates the onset of the item. Hippocampal spectrograms include data from any electrodes in CA1, CA2, CA3, dentate gyrus or subiculum (left $N = 26$, right $N = 21$). Lateral temporal lobe electrodes include any electrodes within 2.5 cm of the coordinate of the region that showed the strongest memory-related effect in the 1–10 Hz band (left $N = 31$, right $N = 30$). **b** Top panels show example individual electrode power spectra for recalled (red) and not recalled (blue) items using average power over the 0–1500 ms item presentation interval. Shaded regions indicate ±1 SEM. Bottom panels show t-statistics resulting from two-sample t-tests comparing z-scored power between recalled and not-recalled items at each frequency. Colored regions indicate significance at $p < 0.05$. From left to right: subjects 47, 42, 29, and 18

paired-sample t-test, $t(49) = 5.2$, $p < 10^{-6}$, see Methods). However, this boundary-related performance boost did not seem related to the left hippocampal low-theta memory effect because the magnitude of this effect was similar when separately calculated for items both near and far from boundaries (paired-sample t-test, $p > 0.2$).

**Comparing memory and navigation-related oscillatory activity.** Research on hippocampal theta has usually been focused in two domains: memory and navigation. Often these two processes are examined in separate experimental paradigms[9,32], making direct comparisons between theta's role in each domain difficult. To compare memory-related and navigation-related neural oscillations, we examined the timecourse of hippocampal activity throughout the task, which included not only the memory-encoding periods mentioned above but also navigation periods without memory demands. These navigation periods represent epochs when the subject was fully in control of their movement in the environment, with the goal of quickly reaching the target treasure chest.

To provide a baseline for characterizing hippocampal signals related to navigational movement and memory, we normalized the power at each electrode relative to the pre-trial baseline, in which patients were still in the virtual environment but had not yet initiated the trial. We then measured the power at each electrode when patients were navigating but not viewing study items (labeled Nav), as well as measuring power over the timecourse of memory encoding. Finally, we examined control events which occurred when a chest opened but was empty, and thus no encoding took place (No Item).

This approach allowed us to assess the absolute levels of oscillatory power for different behaviors (Fig. 5). At 1–3 Hz, there

was an increase in power following the onset of item presentation in both the left and right hippocampi. In the left hippocampus the magnitude of this increase was greater for recalled items (red) vs. non-recalled items (blue) in the −0.6 to 1.7-s interval ($p$'s < .05, paired-sample t-tests at each time bin; nonparametric clustering[33]), consistent with the results shown earlier. A different effect was present in the right hippocampus. Here, as before, there were no significant clusters of memory-modulated activity. However, 1–3-Hz power during navigation significantly increased relative to baseline (paired-sample t-test, $t(20) = 3.65$, $p < 0.01$), an effect that was not present in the left hippocampus or in any of the other frequency bands. (See Supplementary Figure 3 for a brain-wide analysis of navigation-related activity.)

Although low-theta band power showed strong task-related modulation in these data, we also examined task-related activity at other bands. One notable pattern was that HFA power in the left hippocampus increased during item viewing relative to control empty chests (Fig. 5, top left panel). This dissociation was significant for the 0.4–1.9-s time interval (cluster $p < 0.01$, multiple-comparison corrected). This pattern was also visible qualitatively in the right hippocampus and trended towards significance in the 0.7–1.6-s time interval (cluster $p < 0.1$). Because HFA power correlates with population neuronal spiking[34], this pattern suggests that hippocampal neuronal activity represented the content of a viewed item but does not specifically correlate with memory encoding success. Preliminary recordings of hippocampal single-neuron activity in some patients support this interpretation (Supplementary Figure 4). Together, our results indicate that the primary electrophysiological signature of successful memory encoding in the hippocampus relates to theta-band synchronization rather than the mean rate of neuronal spiking[35].

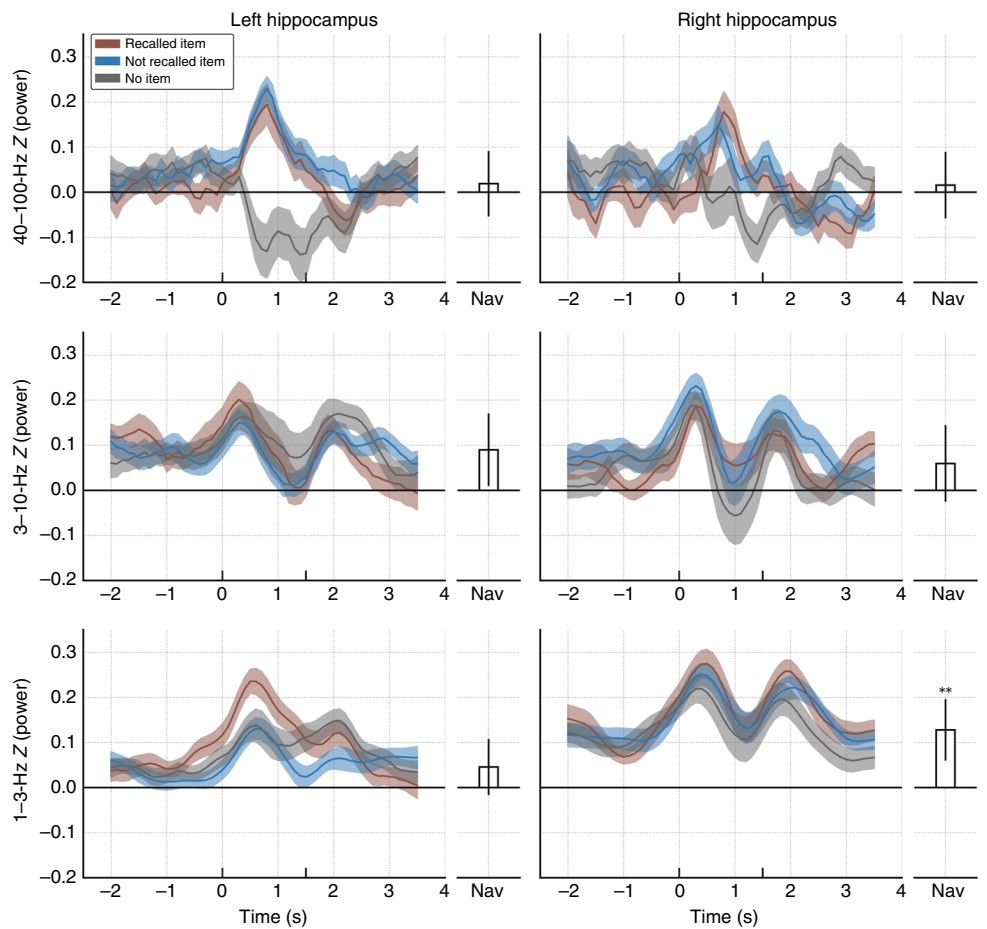

**Fig. 5** Time courses of 1–3-, 3–10-, and 40–100-Hz power before, during, and after item presentation for left ($N = 26$) and right ($N = 21$) hippocampus. All power values are z-scored relative to the pre-trial baseline periods. Red lines indicate power for subsequently recalled items, blue: not recalled items, gray: empty chests (no item). Chests open to reveal an item (or no item for the gray line) at 0 s (onset) and disappear at 1.5 s (offset). Shaded errors regions are ±1 within-subject SEM[70]. The Nav bar represents average power during navigation epochs (periods spent driving to chests), shown with 95% confidence intervals. **$p < 0.01$, Bonferroni corrected

**Lateralization of memory and navigation-related activity**. To determine if there was a reliable lateralization of memory and navigation effects in the hippocampus, we computed memory-related and navigation-related changes in 1–3-Hz power for each subject (Fig. 6). We then performed an ANOVA with two factors: hemisphere (left or right) and condition (memory or navigation). We found no main effects of either hemisphere or condition ($p$'s > 0.4). However, there was a significant interaction ($F(1, 90) = 8.5$, $p < 0.01$), indicating that the memory and navigation effects differed significantly between the hemispheres. We then examined this effect in more detail with post-hoc $t$ tests.

Post-hoc $t$-tests for the navigation condition showed low-theta power was significantly greater during navigation than baseline in the right hemisphere only (right: $t(20) = 3.65$, $p < 0.01$, left: $t(25) = 1.43$, $p > 0.1$). A direct comparison revealed that there was a trend for the navigation effect to be greater in the right hemisphere than in the left (two-sample $t$-test: $t(45) = 1.73$, $p = 0.089$). For the memory condition, low-theta power was significantly greater for remembered items than forgotten items in the left hemisphere only (left: $t(25) = 3.72$, $p < 0.01$, right: $t(20) = 0.01$, $p > 0.1$). A direct comparison revealed that the effect was a significantly greater in the left than in the right (two-sample $t$-test: $t(45) = 2.44$, $p < 0.05$).

We confirmed these findings using a slightly different approach by computing separate two-way ANOVAs for the memory and navigation effects. Here, the factors were hemisphere and either

memory-success or navigation-state. In this framework, the interaction term from the ANOVA can be interpreted as the strength of the lateralization of either memory-related or navigation-related activity. The results of these tests mirrored the findings described above. The hemisphere × memory-success ANOVA resulted in a significant two-way interaction ($F(1, 45) = 5.97$, $p < 0.05$), replicating the above-described $t$-test showing a lateralized low-theta memory effect between the left and right hemispheres. Likewise, the ANOVA with factors of hemisphere and navigation-state again replicated the results of the above-described $t$-test, showing a trend for an interaction ($F(1, 45) = 3.02$, $p < 0.1$). Using this ANOVA framework, we also tested each of these effects in the 3–10 and 40–100-Hz bands. We did not find any significant interactions in any of these four additional tests (all $p$'s > 0.3), indicating that there was no lateralized memory-related or navigation-related hippocampal activity outside of the low-theta band.

**Control analyses**. To rule out the possibility that the laterality effects we observed related to electrodes being placed in abnormal brain tissue, we reanalyzed the data after excluding hippocampal contacts that were ipsilateral to the patient's seizure focus (see Methods). This reduced our dataset to 13 subjects with left hippocampal contacts and 14 with right hippocampal contacts. The results of analyzing this dataset were similar to those described above (see Supplementary Figure 5B). Most notably,

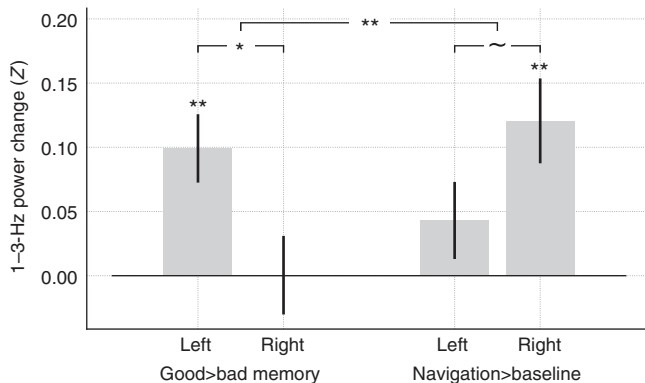

**Fig. 6** Laterality of memory and navigation effects. Task-related 1–3-Hz power changes within left ($N = 26$) and right ($N = 21$) hippocampus, as measured by the difference in mean z-scored power, is shown for the recalled/not recalled contrast and the navigation/baseline contrast. Error bars are ±1 SEM. ˜$p < 0.1$, *$p < 0.05$, **$p < 0.01$. Larger line across all four groups indicates a significant ANOVA interaction ($p < 0.01$)

this reduced dataset continued to demonstrate a significant lateralization of memory-related and navigation-related low-theta activity (ANOVA hemisphere × memory/navigation interaction, $F(1, 50) = 5.5$, $p = 0.02$). These results suggest that our findings of lateralization are not a result of epileptic tissue. Finally, to test whether our effects were due to inter-subject differences, we performed these same analyses using only data from subjects with bilateral hippocampal electrodes ($N = 10$). While the hemisphere × condition interaction was no longer significant ($p = 0.1$) in this limited dataset (likely due to the large reduction in sample size), the general pattern of the effect was retained (see Supplementary Figure 5C).

To determine whether the signals we observed reflected narrowband oscillations as commonly found in the rodent hippocampus, we performed an analysis to specifically identify narrowband oscillations by distinguishing them from the background power spectrum[34,36]. This analysis demonstrates that the left hippocampus reliably exhibits ~3-Hz narrowband oscillations during memory encoding and shows that this pattern is more prevalent during successful than unsuccessful memory formation (Supplementary Figure 6). In contrast, this analysis did not clearly show that the navigation-related activity in the right hemisphere we measured was narrowband, perhaps indicating that this navigation-related activity exhibits wide frequency variations during the task that prevent it from satisfying our criterion as a narrowband oscillation.

To test whether the lateralization patterns that we observed differed as a function of hemispheric dominance of language function, we separately examined any patient with hippocampal electrodes who showed right-hemisphere dominance for language (according to neuropsychological tests). Only one of 46 patients met these criteria. Results from this patient, who was left-handed and had electrode coverage in the left (but not right) hippocampus, were fully in-line with the group data and showed a strong positive memory-related effect and a weak navigation-related effect. Additionally, removal of this patient from the group analyses did not substantially change the pattern of results.

**Single-item decoding of spatial memory**. We next used a multivariate prediction model to examine the heterogeneity of memory-related electrical signals across the brain. We trained a classifier to predict a patient's memory encoding success based on oscillatory power and compared how classification performance varied as a function of training on differing frequencies and brain

areas. This allowed us to tell whether the spatially distributed neural signals we measured contained independent sources of memory-related information, as the addition of independent information to an existing classifier should improve classification performance.

We used a penalized logistic regression model to distinguish whether a given study item would be remembered or forgotten using either low-frequency features (1–10 Hz), high-frequency features (40–100 Hz), or both. We used the wider 1–10-Hz low-frequency range as a general measure of low-frequency activity in order to account for our finding that the hippocampus and neocortex exhibited memory-related signals at somewhat differing frequencies within this larger range (see Fig. 4a). We measured cross-validated classifier performance for each feature set by computing the area under the ROC curve (AUC) of its output predictions (chance = 0.5). Figure 7a shows the results of this analysis. The mean AUC (across subjects) for a classifier using low-frequency features was 0.555 (one-sample t-test vs. 0.5: $t(49) = 5.90$, $p < 10^{-6}$), compared with 0.527 ($t(49) = 1.88$, $p = 0.06$) for a classifier trained on high-frequency features. We also assessed significance using a shuffling procedure, whereby we computed a null distribution of AUC values based on 100 permutations of each subject's good memory and bad memory labels, and we compared the true mean AUC value to the 100 means derived from shuffled data. For both the low-frequency and high-frequency ranges, the true mean AUC was greater than every shuffled AUC (permutation $p$'s < 0.01). Comparing the low-frequency and high-frequency results directly revealed significantly greater classifier performance for the low-frequency classifier than the high-frequency classifier (paired-sample t-test $t(49) = 2.33$, $p < 0.05$), indicating that the low-frequency components of these recordings contained stronger memory-related signals.

The previous analyses used all of a subject's electrodes for classification. To distinguish if memory-related neural signals on different electrodes were independent or correlated, we recomputed the AUC as a function of the number of electrodes included in the classification. If information was redundant across electrodes, then we would not expect to see improvements in classifier performance with increasing electrode count. As seen in Fig. 7b, we found that mean classifier performance improved as additional electrodes were added to the model. This suggests, at least in humans, that separate information about spatial memory encoding is present in distributed patterns of oscillations across the brain, rather than being a single unitary signal as indicated from rodent studies.

## Discussion

Using human intracranial recordings, we have shown that successfully forming memories for object–location associations is correlated with increases in the power of low-frequency oscillations. In particular, increased 1–3-Hz (low theta) power in the left, but not right, hippocampus is indicative of successful spatial memory encoding. This effect is accompanied by a broader memory-related power increase at 1–10 Hz in the lateral temporal lobe. Additionally, increased low-theta power in the right, but not left, hippocampus correlates with navigation. These findings provide electrophysiological evidence for lateralized electrical activity in the human MTL that differentiates between spatial navigation and episodic memory. Moreover, our findings suggest that there are distinct task-related low-frequency oscillations between the lateral temporal lobe and the hippocampus.

Our finding of a memory effect that is specific to the left and not right hippocampus lends credence to the view that the two hippocampi are functionally lateralized. This result is consistent with a

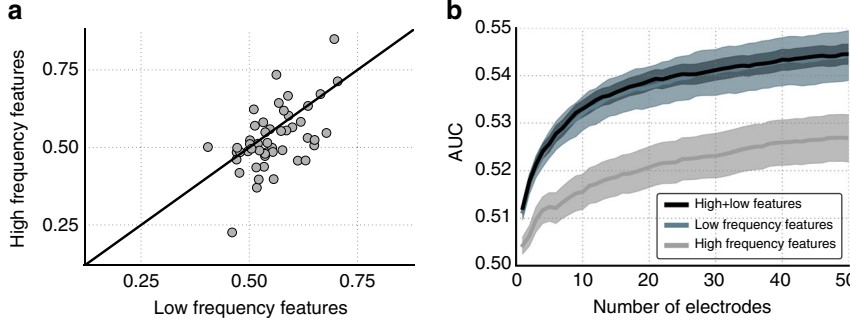

**Fig. 7** Logistic regression classification of subsequent memory success. **a** Area under the ROC curve (AUC) computed using either a classifier trained only low-frequency (1–10 Hz) or high-frequency (40–100 Hz) features. Each point represents the AUC for one subject ($N = 50$). **b** AUC as a function of the number of electrodes used to train the classifier, plotted separately for low-frequency features (blue line), high-frequency features (grey), and combined low-frequency and high-frequency features (black). Shaded errors regions are within-subject standard errors[70]

large body of lateralization findings from lesion and functional imaging studies[20,37–42]. This work indicated that the left hemisphere is associated with episodic memory and that the right hippocampus is linked with spatial processing. In addition, there is some neuromodulation evidence for hippocampal lateralization in rodents (e.g., ref. [43]). However, despite this earlier work, there are a number of reasons to doubt that hippocampal theta oscillations, in particular, would be lateralized. Theta oscillations in rodents are synchronized across both hemispheres[23]. A previous study in humans did not find lateralization of task-related oscillatory activity in the hippocampus[44]. Furthermore, theta oscillations are coupled to entorhinal grid cells[13], which are present in both the left and right hemispheres in humans and rodents[45,46].

Our task included separate intervals for navigation and memory encoding. Although both portions of the task could be considered spatial to some extent, they differed fundamentally in their object–location encoding demands. The navigation phase of the task involved movement and orienting, whereas the item encoding epoch provided a specific time period when patients had to form a new memory for an object–location pair. By separating these phases across distinct time intervals, it likely improved our ability to detect a dissociation between their neural signatures. It is worth noting that our navigation condition, during which translational and rotational movement was under complete control of the subject, consisted of movement for 96% of all timepoints. As such, we did not have adequate data to examine neural signals related to pauses during navigation, when navigational planning may occur.

One critique of virtual reality studies of spatial cognition is that it can be challenging to be sure that a neural signal is specifically related to navigation rather than to any sensory or motor processing inherent to moving through virtual space[47]. Thus, it would be helpful for future virtual reality paradigms to include control conditions to distinguish true neural correlates of navigation from signals related to sensorimotor processes. However, there is evidence that navigation-related hippocampal theta oscillations found during virtual tasks are not the result of low-level sensorimotor processes. For example, Vass et al.[48] showed that movement-related theta oscillations persisted in the absence of motor control or optic flow when subjects were "teleported" across an environment while the screen was blank. This type of result, as well as others[49,50], suggest that our findings are relevant for real-world navigation.

To the extent that theta amplitude indexes hippocampal activation[3], our results suggest that the left hippocampus supports forming a new episodic memory based on associating a new event (or item) to its context. This context may be primarily spatial, as in our task, or it could involve time or other state information, as in other studies that found left-hippocampal activations[51,52]. In

contrast, the right hippocampus may be more focused on purely representing spatial information, perhaps reflecting the activity of place and grid cells in providing a constantly updated representation of a person's current spatial location[45,46]. By revealing the regional heterogeneity of hippocampal theta rhythms, our findings help to resolve discrepancies between two competing views of theta, one concerning theta's role in synaptic plasticity and the other in path integration. Our findings suggest that theta may play both roles simultaneously by distinctly activating each of the hippocampi.

Davachi (2006)[53] showed that successful memory encoding for different stimulus categories was lateralized across the hippocampus, with left activation for verbal memory encoding and bilateral/right activation for pictures and scenes. Unfortunately, our study cannot be directly compared with Davachi's[53] predictions due to our task's design, as we had subjects view words and objects simultaneously during each encoding epoch rather than having different trials with varying presentation forms. Nonetheless, our findings share features with that work in the left hemisphere and suggest a potential link between theta oscillations and fMRI BOLD activity. More fully revealing the potential lateralization of hippocampal oscillations for different stimulus categories will require a different task that includes trials that separately present items from individual categories.

We observed both memory-related and navigation-related power effects at 1–3 Hz in the hippocampus, a lower frequency than the canonical 4–8-Hz theta range identified with scalp EEG data, and consistent with previous studies implicating this lower band in navigation and memory[30,54]. Our findings thus build on a body of evidence that behaviorally relevant hippocampal oscillations increase in amplitude at slower frequencies in humans than in rodents ([26,55]; but see ref. [49,50]). Furthermore, a notable counterexample to this trend is a recent study by Crespo-García et al. [54], who reported that decreased, not increased, hippocampal theta during learning was predictive of memory success for object locations in a virtual space. In addition to finding decreased theta activity, Crespo-García et al.[54] found memory effects to be more prominent in the right hippocampus, whereas we found location-memory effects in the left. Though the experimental paradigms were fairly similar, it is conceivable that differences in task design contributed to this discrepancy, perhaps with regards to the nature of the presented stimuli (i.e., 2D pictures in the Crespo-García et al.[54] paradigm vs. 3D objects embedded in the environment in our task) or the timing of the effects. It is notable that the memory-related theta activity in our task was present up to one second before chest opening, in contrast to previous work that showed memory-related effects that followed stimulus onset[54,56,57]. Future work will have to test whether this reflects the unconventional

nature of our spatial task, in which subjects see each chest prior to arrival, or whether human memory-related theta oscillations simply appear on a wider timescale so that significant effects are apparent prestimulus, as seen in an earlier study[58].

Theta power has been reported to both increase and decrease during item encoding (see Hanslmayr and Staudigl[59] for an overview). In verbal memory tasks such as free recall or paired-associate learning of word lists, theta power decreases are often reported for good memory[56,60], but the direction of this effect may change depending on the exact time period analyzed[58], or even, intriguingly, as a function of within-task manipulations of testing conditions[57]. Thus, in light of these earlier studies on the electrophysiology of human memory, it might be considered surprising that we found that the power of low-frequency signals positively correlated with memory encoding and, inversely, that the power of HFA tended to correlate negatively. We hypothesize that these differences reflect the spatial demands in our task, such that low-frequency signals across the brain are more relevant during spatial processing, whereas memory tasks that are purely verbal rely on brain networks that correlate with HFA activations. Our findings on the role of low-frequency oscillations in spatial memory are consistent with earlier work in rodents showing that hippocampal theta power positively correlated with spatial memory encoding[4], and they build on previous noninvasive measurements from magnetoencephalography, which showed hippocampal theta positively correlated with performance in virtual navigation[28,61].

In conclusion, we have shown evidence for a dissociable contribution of human hippocampal oscillations to different aspects of hippocampus-dependent behaviors, finding correlates of encoding success in the left hippocampus and correlates of navigation in the right hippocampus. In addition to informing our fundamental understanding of how the brain supports cognition, our results could be useful translationally by providing spatial and electrophysiological targets for using neuromodulation to enhance memory or spatial functions[62].

## Methods

**Participants.** Forty-six patients (28 male, mean age: 31.1 years) with medication-resistant epilepsy performed our TH spatial-memory task. Patients were implanted with subdural electrodes on the cortical surface, and/or depth electrodes extending into deeper brain structures. Four patients underwent secondary implantations or had their electrode montages changed and thus are counted as additional subjects, for a total of 50 unique patient–electrode configurations. Electrode placement was determined solely by the clinical team at each collaborating hospital for the purpose of localizing seizure foci. Data were collected at Thomas Jefferson University Hospital (Philadelphia, PA), Mayo Clinic (Rochester, NY), Hospital of the University of Pennsylvania (Philadelphia, PA), Emory University Hospital (Atlanta, GA), University of Texas Southwestern Medical Center (Dallas, TX), Dartmouth-Hitchcock Medical Center (Lebanon, NH), and Columbia University Medical Center (New York, NY). The research protocol was approved by the institutional review board at each hospital, and informed consent was obtained from each participant.

**Experimental task.** Patients performed the TH spatial-memory task in which they navigate a 3D beach with the goal of remembering the locations of encountered objects. The task was developed using Unity3D. Patients control their movement using a handheld joystick. The virtual beach ($100 \times 70$ virtual units, 1.42:1 aspect ratio) is bounded by a wooden fence on each side. Each session is comprised of 40 trials. Subjects completed an average of two sessions.

On each trial, the subject is placed on the ground at the edge of the environment, in either the hut or the semicircle of totem poles (see Fig. 1 for an overhead view of the environment). Subjects remain at this location until they initiate the trial with a button press, and then they navigate to the visible treasure chest. Subjects have full command of their movements during these navigation periods and control translational and rotational movement via a joystick on a handheld game controller. Subjects are encouraged to travel to the target chest as quickly as possible and receive bonus points for efficient navigation. Upon arrival at a chest, subjects are automatically rotated to directly face the chest, and the chest then opens to either reveal a common object or an empty chest. After 1500 ms, the chest (and item if present) vanish. Subjects travel to four chests during the course

of a trial, with either two or three containing an object (Fig. 1a, inset 1 and 2). Because the number of chests containing an object randomly varied between two and three, subjects could not predict whether the current target chest contained an object. This served to remove effects of expectation and to encourage subjects to always attend to their current location as they approached a chest. Subjects encounter 160 chests in a full session, 100 of which contain objects and 60 of which are empty (either one or two empty chests per trial). See Supplementary Movie 1 for a video of patient 1 doing a single trial of the task.

After traveling to the fourth chest, the subject is smoothly moved to one of the ends of the beach where they have an elevated 3/4 overhead perspective view of the environment. The reason for this perspective shift was to speed the retrieval period, conserving patient testing time to provide a relatively larger number of memory encoding events. Additionally, the retrieval location randomly alternated between two ends of the environment to encourage the use of an allocentric strategy. Subjects then play a distractor minigame (Fig. 1a, inset 3), where they must track which of three constantly rearranging boxes contains a coin, where the box positions are randomly swapped four times. After the distractor game, the response phase of the trial begins. Here, subjects are shown a randomly selected object from that trial (Fig. 1a, inset 4), and on-screen text asks "Do you remember where to find the <object>?". Subjects then select their memory confidence from the choices "Yes", "Maybe", or "No". After this confidence response, subjects use the joystick to control a target circle (radius 13 virtual units) that is placed on the beach, and they indicate the location in the environment where they believe the object had been encountered. After being probed for all the objects from that trial, they are given feedback. The patient is told that a given response is considered correct if the patient placed the target circle so that it included the respective object's true location. After each trial patients gain or lose points according to their response correctness and confidence rating.

The confidence rating was designed to promote engagement with the task and provided a secondary metric of performance. If a subject indicated they did not know where the item was located ("No"), then only 50 points were gained if the object location was correctly identified, and no points were lost if not. If a subject indicated medium confidence ("Maybe"), then 100 points were gained if the object location was correctly identified, and 50 were lost if not. If a subject indicated high confidence ("Yes"), then 200 points were gained if the object location was correctly identified, and 350 were lost if not.

To rule out the possibility that our neural analyses of subsequent memory could be influenced by variability in the memorability of the objects in our pool, we analyzed mean recall performance as a function of object identity. We performed a one-way repeated-measures ANOVA with object identity as a within-subject factor to determine if there was a reliable difference in performance from item to item. We did not find significant effect of object identity ($F(100, 4900) = 1.09$, $p > .2$), leading us to conclude that variability in the recall rates for individual objects was not strong enough to drive the observed neural memory effect.

**Intracranial recordings.** As each patient performed our task, intracranial EEG (iEEG) was recorded using Nihon Kohden EEG-1200, Natus XLTek EMU 128 310, or Grass Aura-LTM64 systems, sampled between 500 and 2000 Hz. Signals were initially referenced to a common contact placed either intracranially, on the scalp, or the mastoid process.

To reduce confounding noise artifacts, we used a bipolar referencing scheme whereby we identified all pairs of immediately adjacent electrodes and calculated the voltage difference between both contacts in the pair[56]. The location of these new virtual electrodes was taken to be the midpoint between the two physical contacts (inter-electrode spacing: 10 mm). Data from the virtual electrodes were used in all analyses and are referred to as in the text as "electrodes". To further reduce electrical line noise, a band-stop 4th order Butterworth filter was applied at 58–62 Hz. To eliminate events contaminated by epileptiform activity, we excluded time periods of interest if the kurtosis of the voltage trace exceeded a threshold of 5[63]. This resulted in the exclusion of an average of 7.1% ± 1.7% of experimental events.

Additionally, clinicians at each collaborating hospital identified which electrodes were located in seizure onset zones. For our control analysis that only included a subset of hippocampal contacts, we excluded a subject's hippocampal electrodes in a given hemisphere if any MTL electrode in the ipsilateral hemisphere fell inside of a clinically defined seizure onset zone.

**Anatomical localizations.** Post-implant CT images were coregistered with pre-surgical T1 and T2 weighted structural MRIs with advanced normalization tools[64]. For patients with MTL depth electrodes, hippocampal subfields and MTL cortices were automatically labeled in a pre-implant, T2-weighted MRI using the automatic segmentation of hippocampal subfields (ASHS) multi-atlas segmentation method[65]. Subdural electrodes were localized by reconstructing whole-brain cortical surfaces from pre-implant T1-weighted MRIs using Freesurfer and snapping electrode centroids to the cortical surface using an energy minimization algorithm to account for possible distortion or brain shift. Each subject's T1-weighted MRI was additionally registered to an average T1 constructed from a sample of 101 patients[66], facilitating group-level comparisons of subdural electrodes on the cortical surface. For analyses of hippocampal electrodes, we included electrodes in CA1, CA2, CA3, dentate gyrus, and subiculum. For cortical region of interest (ROI) analyses, we used Freesurfer to label electrodes based on the Desikan–Killiany brain

atlas[67]. Electrodes included in the temporal lobe panel of Fig. 4 were taken from superior temporal, middle temporal, and inferior temporal regions.

**Spectral analyses**. To analyze the spectral properties of the recorded signals, we calculated the continuous Morlet wavelet transform (wave number 5) at 50 logarithmically spaced frequencies between 1 and 200 Hz. For analyses of item encoding (Figs. 3, 6), spectral power was computed at each sample in the 0–1500-ms item presentation window and then averaged over time. For analyses of average power during navigation and non-navigation periods (Fig. 6), power was computed and then averaged over each variable duration navigation and pre-trial baseline period (see "Analysis of navigation epochs" below). A 3000-ms buffer was added to both ends of all power computations before wavelet decomposition in order to minimize edge effects. The resulting power values were then log-transformed (with the exception of the individual electrode power spectra shown in Fig. 4b, we $z$-transformed the log-power values within session). This $z$ transformation was performed separately for every session, electrode, and frequency, by subtracting the mean log-power across all event types (encoding, navigation, and baseline) and dividing by the standard deviation.

For time–frequency spectrograms (Fig. 4), we averaged log-power into 69 overlapping windows of 100 ms each in steps of 50 ms, between −1500 and 2000 ms relative to item onset. We then $z$-scored as described above, now separately for session, electrode, frequency, and timepoint. For the longer time periods shown in Fig. 5, we averaged log-power into 56 overlapping windows of 500 ms each, in steps of 100 ms, between −2250 and 3750 ms relative to item onset, and we normalized the $z$-power values based on the mean power of the baseline condition.

To determine whether changes in spectral power were due to the presence of narrowband oscillations, we performed the oscillation detection procedure of Manning et al.[34]. First, we computed the mean power spectra between 1 and 50 Hz for each electrode and condition and used a robust regression to fit the $1/f$ background power spectrum. We labeled any frequency where the residual power was greater than one standard deviation above the background $1/f$ as exhibiting narrowband oscillatory activity.

**Behavioral performance**. We measured performance on each trial by first computing the raw error distance for each response. This is computed as the Euclidean distance between the item's actual location and the position where the subject placed the target response circle. Given the boundaries in our environment, the distribution of possible error distances is greater for items with locations near boundaries compared to objects in the center of the environment. To normalize each object to the same response range, we transformed the raw error distance for each response to an accuracy score. The accuracy score is computed as the percentile rank of the actual Euclidean distance error relative to all possible distance errors that could have been made, given the item's location relative to boundaries. An accuracy value of 1 corresponds to a perfect response and 0 is the worst possible response. This ranking procedure thus ensures that the distributions of possible scores are identical for all object locations.

We identified an accuracy threshold for each patient, which split their data into two balanced categories, remembered and forgotten. This threshold was computed as the median accuracy across all of a patient's responses from all sessions. In order for a given response to be considered remembered, the accuracy must have been greater than the median threshold and, additionally, the subject must have selected either the medium or high confidence response. Note that this refined performance measure is slightly different compared to the simple method used to reward the patient with points during the task.

We compared behavioral accuracy between items learned in different spatial locations. In one binning scheme, we divided the environment into an inner region and a boundary region, where the inner region was a rectangle with the same aspect ratio as the whole environment and comprised of half of the total area, and the boundary region was the remaining outer area[68,69]. In another binning scheme, we divided items into near and far items based on their distance to the testing location. Here, near items were encountered in the near half of the field relative to the testing location and far items were encountered in the far half.

**Subsequent memory analyses**. To identify memory-related signals, at every electrode, frequency, and time bin, we computed the difference in mean $z$-scored power between subsequently remembered and subsequently forgotten items. The resulting difference represents the degree of memory-related change in spectral power for a given electrode, frequency, and time, where positive values indicate greater power for remembered items.

To create the brain maps in Fig. 3, we used an average brain surface that was created based on T1-weighted MRIs from 101 patients from a different dataset[66]. All of the electrodes in our dataset were registered to this average brain in MNI space, allowing for group analyses to be represented on this average surface, as well as on cross sections of the hippocampus. For every electrode, $z$-score differences were first averaged within a frequency range of interest (between 1 and 3 Hz, 3 and 10 Hz, and 40 and 100 Hz). We then found every vertex on the average surface that fell within 12.5 mm of an electrode's coordinates, and we tagged each of these vertices with the electrode's change in $z$-score. For contacts localized to the

hippocampus, we used a distance threshold of 3 mm to tag all nearby voxels. Values at each vertex or voxel were then averaged across electrodes within each patient to create patient-specific maps. Patient-specific $z$-score maps of hippocampal activity were smoothed with a 4 mm Gaussian filter before subsequent analysis. We performed a one-sample $t$-test at each location comparing the distribution of subject $z$-score differences to zero. To determine significance thresholds, we computed a null distribution of $t$-statistics by randomly sign-flipping half of the values at each vertex for each subject and recomputing the group-level $t$-statistics, and performed this procedure 1000 times. We identified negative and positive thresholds as the 2.5th and 97.5th percentiles of this null distribution. Brain coordinates with less than five subjects were excluded from analyses and colored as black in the figures. Shaded red and blue regions indicate the $t$-statistic resulting from the one-sample $t$-test at that vertex.

To create time–frequency plots (Fig. 4), we averaged changes in $z$-scores within each ROI, such that each patient who had electrodes in a given ROI contributed one observation per frequency and time bin. These patient-level $z$-score differences were then averaged across patients to compute the group-level effects. Then, for each frequency and time bin, the distribution of $z$-score differences was compared to zero using a one-sample $t$-test. Areas of significance are outlined in black and were determined using cluster-based non-parametric statistics that controlled for multiple comparisons[33].

**Analysis of navigation epochs**. During each trial, we can identify navigation epochs. Navigation epochs are variable in length according to the patient's driving decisions and are comprised of the four time periods per trial spent navigating to the current target chest. We contrast navigation epochs with baseline epochs, which are the time periods between when the subject is placed on the beach and when the subject presses a button to initiate each trial. The navigation/baseline contrasts shown in Figs. 5 and 6 follow the same data aggregation and statistics as described for the memory analyses.

**Multivariate classification**. We used a L2-regularized logistic regression classifier to predict memory encoding success for each individual item on the basis of the patient's neural signals. We computed spectral power for each electrode following the above methods, except now using 10 log-spaced frequencies between 1 and 10 Hz (low frequencies) and 10 log-spaced frequencies between 40 and 100 Hz (high frequencies), averaged across the 0–1500 ms item encoding period. We computed power in this manner in order to have a matched number of frequency features for both frequency ranges. Thus, for each subject, the number of classifier features was equal to the number of electrodes by the number of frequencies. The ability of a subject's classifier to discriminate subsequently recalled from forgotten items was measured by the area under the receiver operator characteristic curve (AUC). AUC was computed using a leave-one-session-out cross validation scheme for subjects with multiple sessions and a leave-one-trial-out scheme for subjects with only one session of data.

To compute AUC as a function of number of electrodes (Fig. 7b), we randomly selected one half of a subject's encoding observations, and we performed two-sample $t$-tests at each input feature between subsequently recalled and not-recalled items. We then found the maximum absolute value of the resulting $t$-statistics across frequencies, and we sorted the electrodes in order of descending $t$-statistics. For $N \in 1.50$, we trained our classifier on the first $N$ electrodes, and we tested on the held-out half of the data and computed the AUC. We then repeated this procedure 100 times and computed the mean AUC for each $N$. We performed this procedure using only low frequencies, only high frequencies, and low and high frequencies together.

**Code availability**. Data analyses were performed with custom Python code. Analysis code is available upon request to the corresponding author.

**Data availability**. All de-identified raw data may be downloaded from http://memory.psych.upenn.edu/RAM.

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

## Acknowledgements

The authors acknowledge support from the DARPA Restoring Active Memory (RAM) program (Co-operative Agreement N66001-14-2-4032) and NIH Grants MH104606 & 1S10OD018211-01. The views, opinions and/or findings expressed are those of the author and should not be interpreted as representing the official views or policies of the Department of Defense or the U.S. Government. We thank Blackrock Microsystems and Medtronic, Inc. for providing neural recording equipment. We are indebted to all patients who volunteered their time to participate in our study.

## Author contributions

J.M. and J.J. wrote the manuscript. J.M. and M.T. analyzed the data. J.M., A.J.W., S.A.L., and J.J. designed the experiment. M.T., S.A.L., S.A.S., C.A.S., E.H.S., M.R.S., A.S., A.A.A., G.A.W., S.M., C.I., K.A.D., B.L., and P.A.W. performed data collection and recording. S.R.D., J.M.S., and R.G. processed the neuroimaging data. All authors edited the manuscript.

## Additional information

**Competing interests:** The authors declare no competing interests.

