## [Peer Review File · Nature Communications]

Reviewers' comments:

Reviewer #1 (Remarks to the Author):

The authors examine changes in oscillatory power in low theta (1-3Hz), theta (3-10Hz) and gamma (40-100Hz) bands on subdural electrode grids and hippocampal depth electrodes in epilepsy patients during the encoding of object-location associations and spatial navigation as part of a desktop VR task. First, they demonstrate that low theta power in the left - but not the right - hippocampus, and a broader increase in 1-10Hz power in the left lateral anterior temporal lobe, are associated with successful memory encoding. Second, they show that low theta power in the right hippocampus is increased during navigation compared to a pre-trial baseline period. Finally, they show that subsequent memory success on a trial by trial basis can be decoded from 1-10Hz power across electrode contacts.

These findings are interesting and add to a growing body of literature regarding the role of low frequency oscillations in human memory and spatial navigation function. The paper is generally well written and I believe it will be of significant interest to researchers in the field. However, I feel that some detail is lacking and several additional analyses must be performed to fully substantiate the author's claims. In particular, the statistical analyses used are often insufficient to fully support the conclusions drawn and, in some cases, simply incorrect. Each of these points is described in more detail below.

Major Points

Results: A little more information on the patient population and pre-processing of intracranial EEG recordings in the main body of the manuscript would be useful. In particular, how many patients had depth electrodes implanted in a single hemisphere, and how many had bilateral depth electrode implants? In addition, I can find no mention of any attempt to identify or exclude artefact trials, but this is common practise in both EEG and MEG studies. Indeed, this procedure may have greater importance here, where recordings can presumably be made from pathological tissue. Finally, in the interests of generalising these results to a healthy population, it would be useful to establish whether the main findings differ between patients according to clinical criteria, such as the presence of hippocampal sclerosis on a medial temporal lobe seizure onset zone, if this data is available.

Results: While I appreciate that the choice of frequency bands here is driven by the results of previous human intracranial recording studies, the authors could make a little more effort to ascertain whether there is any genuine oscillatory activity within each frequency band during both memory encoding and navigation (see comment below) by showing appropriate power spectra. They currently show time frequency plots in Figure 4a, but I would prefer that Figure 4b also showed separate group level power spectra for successful v unsuccessful memory encoding on left and right hippocampal depth electrodes, rather than examples from individual subjects from which no meaningful conclusions can be drawn. In addition, it would seem more sensible to accompany these with group level time-frequency plots and power spectra for the specific anterior temporal lobe regions that show prominent

subsequent memory effects in Figure 3, rather than data averaged across all lateral temporal lobe electrode contacts.

Results: I am not sure what Figure 5 adds to the manuscript. As far as I can tell, these plots show the time course of power changes in each frequency band across the time window around memory encoding, but this data has already been shown in Figure 4a. I would rather see a summary of the navigation v baseline data presented in the same style as Figure 3, with glass brain images to show changes in each frequency band during navigation across all sampled regions, and summary bar charts showing the change in each frequency band on depth electrode contacts in left and right hippocampus, respectively. As mentioned above, summary power spectra – to ascertain whether specific oscillatory peaks appear during navigation v baseline – should also be included. Some of this data is currently present in Figure 5, but the comparison with memory encoding results – which is the main focus of this manuscript – would be greatly facilitated by presenting these data in the same format. I realise this would occlude the results of ‘control’ events but, aside from the differences in gamma power during memory encoding, I am not sure these results are relevant to the main findings of the manuscript anyway. If the authors do wish to include them, they should do so in a separate figure.

Results: More clarification is required, in both the Results and Methods, as to what constitutes a ‘navigation’ epoch – in particular, whether these epochs incorporate both movement and stationary periods. Do participants automatically move forward in the environment, once the trial is initiated, and simply steer left and right with the joystick, or can they also make self-directed forward and backward movements? If the navigation epochs include both movement and stationary periods, then all comparisons of oscillatory power between navigation and pre-trial baseline periods should be replaced with comparisons of oscillatory power between movement and stationary periods within each navigation epoch, to facilitate comparison with both previous human intracranial recording studies and the rodent literature. Finally, the authors might perform some control analyses to ensure that changes in oscillatory power observed during navigation do not simply arise from motor control of a joystick or, if this is not possible, at least acknowledge in the Discussion that this confound exists.

Results: I am not convinced that the appropriate statistics have been used to establish the laterality of memory encoding and navigation effects.

First, it is not clear to me whether recalled > not recalled and navigation > baseline are equivalent conditions, and therefore whether including them in the single ANOVA described on page 8 (with results shown in Figure 6) is valid. I am happy for the authors to keep this analysis in the manuscript, but it should be supported by additional statistics. In particular, the authors should include separate ANOVAs to assess the laterality of theta oscillations related to memory and navigation: i.e. one ANOVA with factors of hemisphere (left or right) and memory condition (recalled or not recalled); and one ANOVA with factors of hemisphere (left or right) and navigation condition (navigation or pre-trial baseline). In each case, laterality would be supported by a significant interaction between hemisphere and condition.

Second, to further support these results, the authors should assess whether power values are different from zero in each condition and hemisphere – i.e. it appears from Figure 6 that low theta power for recalled > not recalled is only greater than zero in the left hemisphere, but theta power for navigation v baseline is significantly greater than zero in both hemispheres. If this is the case, then the authors cannot claim that navigation is associated with increased low theta power in the right hippocampus only, rather that the magnitude of low theta power increases during navigation are (marginally) larger in the right hippocampus. This is a subtle but important distinction.

Third, it is not clear to me why the authors show average t-statistics in Figures 3 and 6, nor is this correct. These are group level analyses, and therefore a single group level t-test should be performed on average z-scored power values (or average difference in power between conditions) across patients. The authors must not perform statistics on statistics and incorporate differences in within-subject variance into between-subjects analyses. Hence, they must change Figures 3 and 6 to show average power changes across subjects, and report the outcome of a single t-test on those power values across subjects at each cortical location / for each contrast.

Fourth, I am assuming that the ANOVA described on page 8 (with results shown in Figure 6) treats data from left and right hippocampi as independent samples, given that they can be drawn from different patients. However, claims of laterality might be further supported by repeated measures comparisons of data from each hemisphere in the same patient, where patients with bilateral depth electrode implants are available. This analysis should also be included.

Fifth, it would be useful to mention, at least briefly, the results of similar analyses on the 3-10Hz theta and 40-100Hz gamma bands, to further ascertain the specificity of the reported effects to the low theta band. In fact, data from all three frequency bands could be included as an additional factor in each of the ANOVAs described above, to establish whether there are any main effects of, or interactions with, frequency band.

Finally, the authors state that "...navigation-related low-theta power was greater in the right hemisphere than the left ($t(45) = 1.91, p = 0:06$)", but their statistical analysis clearly does not support this conclusion. This sentence must therefore be moderated to acknowledge that this is a weak and non-significant effect.

Results: It is not clear to me why the authors lump both theta bands together in their decoding analyses, given that the vast majority of significant results mentioned above are specific to the low theta (1-3Hz) band. If successful discrimination cannot be supported by power changes in the low theta band only, then this should be mentioned as an important caveat to the preceding findings. Why not perform this analysis with each frequency band separately?

Minor Points

Abstract: "Brain signals" seems a little simplistic for a scientific journal. I suggest "local field potential", "EEG activity" or similar

Introduction: "The theta oscillation in rodents is specifically hypothesized to be involved in memory because its amplitude correlates with memory encoding [Lega et al., 2012, Seager et al., 2002]" – neither of these citations describe rodent studies. The former examines theta activity in humans, the latter in rabbits. Hence, this sentence is misleading. For studies linking rodent theta activity with memory performance, I suggest Winson (1978) and / or McNaughton et al. (2006):

Winson J (1978) Loss of hippocampal theta rhythm results in spatial memory deficit in the rat. *Science* 201: 160-163

McNaughton N, Ruan M, Woodnorth M-A (2006). Restoring theta-like rhythmicity in rats restores initial learning in the Morris water maze. *Hippocampus* 16: 1102-1110

Introduction: "The statistical power of our study is greater than earlier work of this type [Miller et al., 2013, Suthana et al., 2012, Jacobs et al., 2016] because we enrolled 46 subjects and because each task session included 100 memory encoding events" – Again, I find this sentence a little odd, as none of the cited studies assess theta activity during memory encoding or spatial navigation, so I am not convinced that they constitute "earlier work of this type". Presumably, a better comparison would be made with earlier human intracranial studies of theta activity during navigation, which generally have smaller sample sizes. However, I also note that several earlier human intracranial studies of theta activity during memory function use similar or much larger sample sizes (i.e. Lega et al. 2011, Burke et al. 2014).

Methods: "46 patients (28 male, mean age: 31.1 years) patients" – 'patients' is repeated

Methods: "Here subjects are shown a randomly selected object from that trial (Figure 1A, inset 4) and asked to move a target circle to indicate the location in the environment where it had been encountered. Subjects indicate their memory confidence on a three-level scale (Yes/Maybe/No), and then, from the same vantage point, use the joystick to move a target circle (radius 13 virtual units) to where they think the item was located (Figure 1A, inset 5)" – I am confused, are the participants asked to indicate the remembered location of an object before or after the confidence judgement, or both?

Methods: "Power values at each frequency were z-transformed within session" – to be clear, was the z-scoring performed across all epochs (i.e. pre-trial, memory encoding, and navigation) on the same electrode within each session? Please provide some more details here.

Methods: "Euclidean" should be capitalized.

Reviewer #2 (Remarks to the Author):

Miller et al. present an impressive study of hippocampal oscillations underlying human

spatial memory and navigation. The study is robust in the number of subjects recruited. Its novelty relates mainly to the lateralization aspect. In the left hippocampus, the amplitude of oscillations in the 1–3-Hz “low theta” band increased when viewing subsequently remembered object–location pairs relative to those that were subsequently forgotten. In the right hippocampus, low-theta activity increased during periods of navigation.

The relationship of theta oscillations to success of encoding in humans is not new, but the lateralization issue is a significant addition. At the same time, the main novelty of the paper, the lateralization, may be confounded by lateralization of underlying epilepsy and its physiological concomitants

While interesting, I am not sure that these findings present the level of novelty and wide readership interest appropriate for Nature Communication. With appropriate revisions, it might be of interest to a more specialized journal. However, there are significant concerns as stated below.

1) Despite authors’ claims, the treasure hunt task does not build on the classic water maze and in fact, is different from that task. Specifically, the utilized paradigm is a guided navigation task and subjects move towards visible targets, as opposed to finding hidden locations in the water maze. Spatial learning occurs in the form of object-place associations, only when subjects have reached the treasure chests, and not during movement.

2) Indeed, it is a peculiar choice to switch between first-person navigation to a location for the encoding and simply marking the correct location from afar. In the least, some additional analyses are in order, as well as more detailed description of the task. Are items that are located farther from the point of view during testing less accurate than nearby items? Do subjects see the first treasure chest from the testing vantage point but then subsequent treasure chests only from within the beach? This could cause the first item to be more accurately placed due to its greater similarity between training and testing, rather than any memory effect. Were subjects ever tested on the location of empty chests; were they informed that they would not have to remember the locations of those items?

3) As stated above A critical issue pertains to lateralization. When considering lateralization of function as studied in epilepsy patients, it's important to consider the lateralization of seizure onset. I'd like to see a supplementary figure showing that the results hold up, even when limiting analysis to electrodes that are fully outside of the seizure onset zone. The methods should also describe how interictal baseline activity was treated when analyzing the physiology.

•

4) In their localization figure, Figure 2B, it seems that their automatic segmentation algorithm has not worked well. It appears that there are areas shown as subiculum where there supposed to be Dentate gyrus, CA2 and CA3 areas.

5) Regarding statistics:

a) Please report Ns and other relevant statistics in your figure legends.

b) Additionally, why is using t-statistic justified? Are the distributions normal?

c) In figure 6 bar plots, it would be nice to see individual data points as well as box plots.

d) Statistics in Figure 4: It is not statistically appropriate to calculate the mean difference of z-scored distributions (in this case power). The authors might have to think about a better way to normalize their data as opposed to z-scoring. Also, please include raw traces from

several trials showing the oscillations.

e) Using a median-split within patient to define remembered vs forgotten doesn't seem fair, as some patients likely remembered most items and others might well have forgotten most. Perhaps an across-subject split would be better. If the across patient accuracy distribution were bimodal and you could split between the peaks, that would be ideal. In either case, please show the distribution and the location of the split

f) Classification: it seems like the main claim of the paper is that the differential contribution of the oscillatory activity to successful recall and navigation occurs in the low theta band. However, in the classification algorithm, the input consists of the 1-10Hz frequency range. If you used low-theta range of 1-3Hz, is the classification algorithm still successful?

g) In the same way that the authors determined the significance of t-stats using the 2.5-97.5%ile of shuffled values, it would be instructive to see the same thing for the AUC analysis instead of a t-test compared to 0.5. Critically, the authors should shuffle the remembered/forgotten labels of the trials without changing the total number to receive each label.

6) Page 5, line 93: "Overall, power in the low-theta and theta bands was significantly greater for recalled items compared to forgotten ones." At least according to Figure 3 (middle row), this statement is not true for theta band, as the values do not pass significance levels.

7) Figure 5: In figure 4, the authors show that in the left hippocampus the difference in power for remembered vs forgotten trials begins half a second before the chest opens. In figure 5, however, the time course for power in the no-item case very closely follows the not-subsequently-remembered trace (in both low-theta and HFA). This is suspicious, as the subjects should not know ahead of time whether a chest will have an item in it, and warrants further exploration/discussion.

8) The description of how "accuracy" was calculated is inadequate. Please include an equation or code, such that the reader could reproduce the same measurement. It is unclear that 0.5 is chance. Perhaps the authors could add a supplementary figure that shows several different overhead maps of the beach with different target locations, and color code the beach by the accuracy that would be assigned if the subject had selected that location.

9) When describing the use of bipolar electrodes, please specify the electrode spacing. Was the location of the "electrode" taken as the midpoint between the two?

10) Supplementary Figure 1: Are all of those 10 neurons recorded from the left hippocampus? If not, how does this support the main claim?

Reviewer #3 (Remarks to the Author):

The manuscript describes a virtual reality experiment in which epileptic patients with depth electrodes had to navigate a beach environment to open chests containing objects and then subsequently, when cued with the items, to re-place it in the locations it was encountered in at learning. This replacement was done from a raised perspective over the environment.

The authors find that the amplitude of low frequency oscillations 1-3Hz in the left hippocampus are higher during encoding when subjects later correctly remember object-

location pairings and the right hippocampal when were higher when the patients were searching for the chests to open them.

This research tackles an important and timely topic – how oscillations in the human hippocampus support navigation and memory. 100s of rodent studies have examined similar questions, but in humans we know little. There is much to commend the authors for in this research. The task is novel and allows multiple aspects of experience to be examined. Research with these patients is challenging so developing a task that will work is not easy and the authors have done a good job with this. The number of patients studied is very high for this type of research which makes the research more useful. The video of the task was very useful for understanding what it would be like for a participant and how clear the distal cues are for navigation.

To improve the manuscript I suggest the authors consider a few points:

1. While navigation epochs do involve navigation, they don't test the type of navigation explored in most studies on the hippocampus – allocentric based navigation to a hidden goal. Rather the authors are tapping into vista-space approach - see e.g. Wiener, J. M., Büchner, S. J., & Hölscher, C. (2009). Taxonomy of human wayfinding tasks: A knowledge-based approach. *Spatial Cognition & Computation*, 9(2), 152-165. Moreover its not clear they are tapping into navigation at all with their contrast to the baseline of not moving, such that it would appear that R-hippocampal 1-3Hz may be driven by visual motion. Given this I think the authors need to be more careful in their framing about what they are referring to as navigation and discuss this issue in more detail.
2. More detail on the subsequent memory effects should be included. It is possible that a certain set of objects are commonly remembered and thus visual stimuli effects are driving the subsequent memory effect rather than a true subsequent memory effect. Similarly, it would be useful to know whether participants were more accurate for items towards the corners of the environment – and if so does this relate to the subsequent memory effect observed?
3. It wasn't clear how often the empty chest occurred – it would be useful when this is introduced to say this frequency. This is important when interpreting the neural responses – i.e. are these novel events?
4. Other articles the authors may wish to cite that provide support their left-sided focus in memory retrieval and specifically object-location process are respectively:
Maguire, E. A., & Frith, C. D. (2003). Lateral asymmetry in the hippocampal response to the remoteness of autobiographical memories. *Journal of Neuroscience*, 23(12), 5302-5307,
Howard, L. R., Kumaran, D., Ólafsdóttir, H. F., & Spiers, H. J. (2011). Double dissociation between hippocampal and parahippocampal responses to object-background context and scene novelty. *Journal of Neuroscience*, 31(14), 5253-5261.

Reviewer #4 (Remarks to the Author):

The authors have presented an insightful paper into the differential oscillatory correlates of

memory encoding and spatial navigation. Human patients with intracranial electrodes completed a video game-style task where they encoded object-location pairs on a virtual beach. Increases in left hippocampal "slow" theta accompanied accurate object-location memory formation, whereas increases in right hippocampal "slow" theta accompanied virtual navigation (relative to pre-trial baseline). These results were interpreted as evidence for a lateralised hippocampus.

The paper is well-written and provides a fascinating insight into the distinction between the left and right hippocampi. However, some issues should be addressed prior to publication.

1) To what extent is the lateralisation differences a result of stimulus type? The encoding task requires verbal processing whereas the navigation task is more visually focused. The second figure in Davachi (2006, *Current Opinion in Neurobiology*) demonstrates a bias in left hippocampal activation for words and right hippocampal activation for objects/faces/scenes. Could the increased left hippocampal theta power for the encoding task reflect the dominance of verbal material, and the increased right hippocampal theta power for navigation reflect the dominance of visual material?

2) Are there sufficient participants with recordings from both the left and the right hippocampus to conduct this analysis within-subjects? A common criticism of lateralisation differences between subjects relates to inter-group differences, but a within-subjects analysis would side-step this issue and would strengthen the claim of a lateralised hippocampus.

3) In figure 4, the differences in oscillatory power do not appear to be event-related, unlike a number of other subsequent memory effects (SME; e.g. Crespo-Garcia et al., 2016, *NeuroImage*; Staudigl and Hanslmayr, 2013, *Current Biology*). Do the authors envision that these findings are analogous to previous findings, or rather a distinct aspect of memory formation? Is the extended SME possibly related to the uniqueness of the task?

4) A number of previous studies have reported broadband increases in high-frequency activity (HFA) relating to successful memory formation, together with low frequency decreases (e.g. Burke et al., 2014, *NeuroImage*; Long & Kahana, 2015, *NeuroImage*). In this report the authors report no significant effects in HFA with subsequent memory. Surprisingly they even report a tendency in the opposite direction. Together with the reported increased low frequency power increases which support episodic memory encoding there appears to be a possibility that the employed task somehow switched the time-frequency pattern associated with memory encoding, i.e. from low frequency power decreases/HFA increases \diamond low frequency power increases/HFA decreases. Can the authors offer an explanation for this apparent discrepancy?

Additional notes:

5) Line 7. "...theta oscillations at 4-8Hz whenever this structure active..." - possibly missing an "is"?

6) Line 67-67 and line 352. A normalised accuracy of 0.5 is referred to as chance here; however I feel that this assumes that the participant selects a random location on the map. If a location was unknown, I imagine that a sensible strategy to give a reasonably accurate response would be to select a location near the centre of the map, rather than at one of the corners, as this is perhaps the closest position to all possible locations. This would bias guessing (and chance performance). Perhaps the "chance" terminology should be dropped.

7) Figure 3. Given the focus on the hippocampus, medial surface plots here would be of interest.

8) Line 349. "we transformed the raw error distance for each response an "accuracy" score...". A word or two may be missing from this sentence.

Overview of the manuscript revisions

We thank the reviewers and editor for their thoughtful comments on our manuscript. We have carefully read and considered these comments while preparing our revisions. Our revised manuscript addresses each and every point that was raised by the reviewers. The changes in the revised manuscript include a number of new data analyses, as well as many edits and clarifications to the text. As a result of making these changes, we feel that the manuscript is dramatically improved.

In the first section below we summarize the biggest changes we made to the manuscript, addressing any points mentioned by the editor or multiple reviewers. The subsequent sections consist of point-by-point responses to individual reviewer comments.

Due to the large number of comments, we have attempted to organize this response letter in a way that is easy to read and understand: red coloring indicates reviewer comments, our responses are indicated in normal black text, indentations indicate new text that we have added to the manuscript. In the revised manuscript, text that has been modified or added is in bold.

Response to comments from the editor

- More specifically, you will see that both Reviewers 1 and 2 are concerned that the lateralization effects you report may be confounded by the epileptic tissue and therefore may not generalize to a healthy population. This concern is one we share editorially, so will only consider a revision if you can convincingly show - by analyzing electrodes outside the seizure onset zone (Reviewer 1 point 1, and Reviewer 2, point 2) - that the lateralization effects are not due to the epileptic tissue.

The Editor and Reviewers 1 and 2 asked us to consider whether our findings related to epileptic brain tissue. To address this issue, as suggested by Reviewers 1 & 2, we reperformed our data analyses while including only signals from electrodes outside the seizure onset zone.

Our main findings continued to be robust with this new method. Response Figure 1 shows the results of this new analysis (right panel), compared with the original analysis (left panel).

Response Figure 1: **Left:** Memory and navigations effects based the original dataset, not excluding any epileptic zones. **Right:** Memory and navigation effects after excluding hippocampi in the hemisphere of seizure onset zones. ~ : $p < 0.1$, * : $p < 0.05$, ** : $p < 0.01$.

As shown in this figure, after excluding the brain sites that were potentially epileptogenic and thus dramatically reducing the size of our dataset, we continue to demonstrate the key findings from the original analysis. Our findings continue to show significant power increases for the memory condition only in the left hippocampus and significant increases for the navigation condition only in the right hippocampus. Importantly, this analysis also continues to show a statistically significant condition-by-hemisphere interaction. Therefore, this analysis provides evidence that our main findings are not related to epileptic activity.

We include this new analysis in our revised manuscript as Supplementary Figure S5. Accordingly, we also have altered the text of the paper to describe and discuss this result, explaining that our findings are not related to features of patients' epilepsy. In the Results on Page 6, we write:

To rule out the possibility that the laterality effects we observed related to electrodes being placed in abnormal brain tissue, we reanalyzed the data after excluding hippocampal contacts that were ipsilateral to the patient's seizure focus (see Methods). This reduced our dataset to 13 subjects with left hippocampal contacts and 14 with right hippocampal contacts. The results of analyzing this dataset were similar to those described above (see Supplementary Figure S5B). Most notably, this reduced dataset continued to demonstrate a significant lateralization of memory- and navigation-related low-theta activity ($F(1, 50) = 5.5, p < 0.05$). These results suggest that our findings of lateralization are not a result of epileptic tissue.

- Additionally, we ask that you do the within-subject comparison suggested by Reviewer 4 to ensure that these results do not just reflect inter-individual variability.

As requested, we now report and discuss the results of a data analysis from the ten patients who had bilateral hippocampal implants. The results of this analysis are shown below in Response Figure 2 and are also now included as part of Supplementary Figure S5. This figure provides support for our main claims of lateralized activity because it largely replicates our primary pattern of results despite reducing the size of our dataset by nearly 75% as a result of excluding subjects without bilateral coverage.

Response Figure 2: Memory and navigation effects for only patients with bilateral hippocampal implants (N=10). ~ : $p < 0.1$, ** : $p < 0.01$. Statistical analyses: increased memory-related activity in the left hippocampus ($t(9) = 1.9, p < 0.1$), navigation-related activity in the right hippocampus ($t(9) = 3.3, p < 0.01$).

In the revised manuscript we now discuss the fact that this result indicates that our lateralization findings are not caused by interindividual differences (Page 6):

Finally, to test whether our effects were due to intersubject differences, we performed these same analyses using only data from subjects with bilateral hippocampal electrodes (N=10). Despite the large reduction in the size of the dataset, we still observed the same general pattern of effects (see Supplementary Figure S5C).

- Finally, it is important to us that you also demonstrate that the lateralization effects are not due to the types of stimuli used, as suggested by Reviewer 4 (point 1), and address all the statistical concerns raised.

Reviewer 4 raised the possibility that the lateralization effects we observed were the result of stimulus type. Specifically, the Reviewer proposed a model in which the left hippocampus preferentially activates for the more verbally dependent encoding period and the right hippocampus preferentially activates for the more visually dependent navigation period. If this were the case, then our conclusion that the left hippocampus is involved in object–location binding in our task and the right hippocampus is not would be premature. Following this logic, the ability to observe a distinction between good and bad memory in a given region would be contingent on the presence of an general stimulus- or category-related response, thus the lack of an effect in the right hippocampus could simply be because the stimulus that was presented didn't elicit an overall activation in this region during encoding.

To address this point, we performed an analysis to determine if the overall levels of hippocampal activation during memory encoding and navigation epochs differed as a function of hemisphere. If we found that the left hippocampus activated more for encoding relative to navigation and that the right hippocampus activated more for navigation relative to encoding, then we would not be able to rule out the possibility that the lateralized memory and navigation effects we observed were simply due to lateralized networks for verbal and visual/scene processing. Therefore, we compared the magnitude of low-theta activity in each hemisphere between encoding and navigation. As shown below in Response Figure 3, this analysis did not show evidence for lateralization of these patterns. A similar pattern is visible in the paper's Figure 5.

Response Figure 3: Change in 1–3 Hz power due to stimulus type (encoding or navigation). This analysis uses all encoding periods and does not distinguish successful from unsuccessful memory.

This analysis shows that although both left and right hippocampi showed increased low-frequency power for encoding relative to navigation, critically, there was no meaningful difference in the size of this effect between the hemispheres. The fact that the left and right hippocampi show similar engagement for memory encoding relative to navigation suggests that both hemispheres are activated in the presence of the verbal stimuli in the task. *Thus, the lack of a memory effect in the right hippocampus is not the result of the right hippocampus simply not being responsive during the memory-encoding phase of the task.* Rather, this result indicates a true difference in the role of each hippocampi with regards to how the perceived information is processed for memory encoding.

Nonetheless, because our task did not vary the categories for item presentation, we cannot entirely rule out the possibility that the exact nature of stimulus presentation had an effect on the observed memory-related activity. However, we do not believe that extrapolating from the findings of Davachi [2006], as mentioned by Reviewer 4, would yield the results we found. Davachi (2006) showed that verbal items elicited memory-related signals in the left hippocampus whereas memory-related activity for objects or scenes was either bilateral or in the right hemisphere. This suggests that we would see memory-related activity in *both hemispheres*, as our task provided multimodal information during encoding (verbal, pictorial, and scene). Instead, because we found only left-sided activation related to memory, it provides a differing result. Further, our result also differs from predictions from animal recordings where both hippocampi show similar oscillations, as we noted in the paper. Finally, we wish to note that the Davachi paper, because it focused only on memory-related contrasts, does not make strong predictions regarding navigation-related activity. Thus, the right-hemisphere navigation signals we describe are a separate phenomenon from what was discussed in that paper.

To summarize these issues, in the revised paper we now note that our results are in part congruent with earlier laterality findings, but that they also differ in important ways (Page 9):

Davachi [2006] showed that successful memory encoding for different stimulus categories was lateralized across the hippocampus, with left activation for verbal memory encoding and bilateral/right activation for pictures and scenes. Unfortunately, our study cannot be directly compared with Davachi [2006]’s predictions due to our task’s design, as we had subjects view words and objects simultaneously during each encoding epoch rather than having different trials with varying presentation forms. Nonetheless, our findings share features with that work in the left hemisphere and suggest a potential link between theta oscillations and fMRI BOLD activity. More fully revealing the potential lateralization of hippocampal oscillations for different stimulus categories will require a different task that includes trials that separately present items from individual categories.

Reviewer 1

- **A little more information on the patient population and pre-processing of intracranial EEG recordings in the main body of the manuscript would be useful. In particular, how many patients had depth electrodes implanted in a single hemisphere, and how many had bilateral depth electrode implants?**

We have added more information on the patient characteristics to the main text. As part of this addition, the revised text now indicates that of the 37 patients with hippocampal electrodes, 27 had electrodes in only one hemisphere and 10 had bilateral implants. As we write on Page 4:

The recordings sampled a range of brain areas, including left and right hippocampi (79 electrodes from 26 subjects and 55 electrodes from 21 subjects, respectively. 10 subjects had bilateral hippocampal implants and 27 had electrodes in only one hemisphere.)

- In addition, I can find no mention of any attempt to identify or exclude artefact trials, but this is common practise in both EEG and MEG studies. Indeed, this procedure may have greater importance here, where recordings can presumably be made from pathological tissue.

As requested, we have revised our data analyses to include artifact rejection at the level of individual trials. As described in the revised Methods (Page 12), we used an established procedure to identify trials with artifacts based on kurtosis, which can identify abnormal EEG as a result of seizure activity, line noise, or other technical issues [van Vugt et al., 2010]. As a result of performing this procedure, 7.1% of all events were excluded from our analyses.

To eliminate events contaminated by epileptiform activity, we excluded time periods of interest if the kurtosis of the voltage trace exceeded a threshold of 5 (Delorme et al., 2007, Sederberg et al., 2007; van Vugt et al. 2010). This resulted in the exclusion of an average of $7.1\% \pm 1.7\%$ of experimental events.

Throughout the revised paper, all Figures and text now describe data analyses that were performed following artifact rejection. None of the key patterns in the manuscript changed significantly as a result of this procedure.

- Finally, in the interests of generalising these results to a healthy population, it would be useful to establish whether the main findings differ between patients according to clinical criteria, such as the presence of hippocampal sclerosis on a medial temporal lobe seizure onset zone, if this data is available.

As described in the preceding section of this letter, the revised manuscript includes a new analysis demonstrating that our findings were not related to epileptiform activity because they persisted after the exclusion of any hippocampal electrodes in the same lobe as the seizure onset zone (Page 6). See above and Response Figure 1.

- While I appreciate that the choice of frequency bands here is driven by the results of previous human intracranial recording studies, the authors could make a little more effort to ascertain whether there is any genuine oscillatory activity within each frequency band during both memory encoding and navigation (see comment below) by showing appropriate power spectra. They currently show time frequency plots in Figure 4a, but I would prefer that Figure 4b also showed separate group level power spectra for successful v unsuccessful memory encoding on left and right hippocampal depth electrodes, rather than examples from individual subjects from which no meaningful conclusions can be drawn.

After consideration, we concluded that the best way to examine this issue is to perform a direct analysis of narrowband oscillations, because averaging power spectra across electrodes does not always produce narrowband peaks for human hippocampal signals because there are variations in theta frequency across individuals, as shown previously [Jacobs, 2014, Zhang and Jacobs, 2015]. Thus, in the revised manuscript we added a new analysis to identify narrowband oscillations, distinguishing these signals from shifts in the background $1/f$ broadband power spectrum. To do this we use methods from Manning et al. [2009], in which we use a robust regression to model each power spectrum and then identify positive deviations above the regression fit as narrowband oscillations. A description of this method appears in our revised Methods as follows:

To determine whether changes in spectral power were due to the presence of narrowband oscillations, we performed the oscillation detection procedure of Manning et al. (2009). First, we computed the mean power spectra between 1–50 Hz for each electrode and condition and then used a robust regression to fit the $1/f$ background power spectrum. We labeled any frequency where the residual power was greater than one standard deviation above the background $1/f$ as exhibiting narrowband oscillatory activity.

The results of this analysis are shown in the new Supplementary Figure S6A, which is duplicated below as Response Figure 4. This analysis shows that we did indeed find sizable counts of electrodes in the left hippocampus that exhibited narrowband oscillations at ~3Hz and that this effect differed in magnitude between good and bad memory conditions. By contrast, this analysis showed mixed evidence that the navigation-related activity we observed in the right hippocampus reflected narrowband oscillations.

Response Figure 4: The percent of electrodes where oscillations were detected for both left and right hippocampi and for memory (A) and navigation (B) conditions. Black horizontal line indicates a significant ($p < 0.05$ difference between conditions, χ^2 test).

The revised text discussing both of these results is as follows:

To determine whether the signals we observed reflected narrowband oscillations as commonly found in the rodent hippocampus, we performed an analysis to specifically identify narrowband oscillations by distinguishing them from the background power spectrum (Manning et al., 2009; Zhang & Jacobs, 2015). This analysis demonstrates that the left hippocampus reliably exhibits ~3-Hz narrowband oscillations during memory encoding and shows that this pattern is more prevalent during successful than unsuccessful memory formation (Supplementary Figure S6). In contrast, this analysis did not clearly show that the navigation-related activity in the right hemisphere we measured was narrowband, perhaps indicating that this navigation-related activity exhibits wide frequency and phase variations during the task that prevent it from satisfying our criterion for being a narrowband oscillation.

- In addition, it would seem more sensible to accompany these with group level time-frequency plots and power spectra for the specific anterior temporal lobe regions that show prominent subsequent memory effects in Figure 3, rather than data averaged across all lateral temporal lobe electrode contacts.

In line with the Reviewer’s suggestion, we have altered our time–frequency plots from Figure 3 to highlight the specific lateral temporal lobe subregions that showed the most prominent memory effects. In the revised manuscript we now show time–frequency plots for the specific subregion in the lateral temporal lobe that we identified as showing the strongest memory-related activity at 1–10 Hz.

After performing the change, the revised figure looks rather similar to the previous version, emphasizing that oscillations in the lateral temporal lobe at 1–10 Hz positively correlate with memory. The updated figure is included below as Response Figure 5, replacing Figure 4A in the manuscript (excluding the inset brain images).

Response Figure 5: Group level time-frequency spectrograms, showing mean difference in normalized power between recalled and not-recalled stimuli. The inset brain images show the 1–10 Hz memory-related changes in power, and the yellow spheres indicate the regions in each lateral temporal lobe with the strongest memory effect. Lateral temporal lobe time-frequency spectrograms include all electrodes within 2.5cm of each sphere.

We have adjusted the caption of the image in the main text to reflect this change:

Hippocampal spectrograms include data from any electrodes in CA1, CA2, CA3, dentate gyrus or subiculum. Lateral temporal lobe electrodes include any electrodes within 2.5 cm of the coordinate of the region that showed the strongest memory-related effect in the 1–10 Hz band.

- I am not sure what Figure 5 adds to the manuscript. As far as I can tell, these plots show the time course of power changes in each frequency band across the time window around memory encoding, but this data has already been shown in Figure 4a.

We agree that it is best to eliminate redundant plots generally. However, we think that Figure 5 is an important component of the paper because it shows the absolute level of oscillatory activity over time for each condition, thus indicating whether the activity at each moment represents an increase or decrease relative to the baseline. By contrast, the mentioned Figure 4A only shows the difference between the two memory conditions without showing the absolute level of such activity—thus, the two figures are complementary.

- I would rather see a summary of the navigation v baseline data presented in the same style as Figure 3, with glass brain images to show changes in each frequency band during navigation across all sampled regions, and summary bar charts showing the change in each frequency band on depth electrode contacts in left and right hippocampus, respectively. As mentioned above, summary power spectra - to ascertain whether specific oscillatory peaks appear during navigation v baseline - should also be included. Some of this data is currently present in Figure 5, but the comparison with memory encoding results - which is the main focus of this manuscript - would be greatly facilitated by presenting these data in the same format. I realise this would occlude the results of 'control' events but, aside from the differences in gamma power during memory encoding, I am not sure these results are relevant to the main findings of the manuscript anyway. If the authors do wish to include them, they should do so in a separate figure.

As requested, we have added a new supplemental figure (Supplementary Figure S3), containing brain plots showing the topography of the navigation-vs-baseline comparison, as well as bar plots of hippocampal effects. In addition, the new Supplementary Figure S6 that we added (see above) is also relevant to this issue because it examines narrowband oscillations during navigation. We preferred this approach rather than removing the existing Figure 5, which we believe is informative.

- More clarification is required, in both the Results and Methods, as to what constitutes a 'navigation' epoch - in particular, whether these epochs incorporate both movement and stationary periods. Do participants automatically move forward in the environment, once the trial is initiated, and simply steer left and right with the joystick, or can they also make self-directed forward and backward movements? If the navigation epochs include both movement and stationary periods, then all comparisons of oscillatory power between navigation and pre-trial baseline periods should be replaced with comparisons of oscillatory power between movement and stationary periods within each navigation epoch, to facilitate comparison with both previous human intracranial recording studies and the rodent literature. Finally, the authors might perform some control analyses to ensure that changes in oscillatory power observed during navigation do not simply arise from motor control of a joystick or, if this is not possible, at least acknowledge in the Discussion that this confound exists.

As the reviewer suggested, we have modified the Methods and Results to more clearly describe how subjects control navigation in the task. Here we explain that during each navigation epoch the subject can fully control movement by either moving forward or backwards, turning by rotating left or right, rotating and moving at the same time, or remaining still. In the Methods, on page 10, we write:

Subjects have full command of their movements during these navigation periods and control translational and rotational movement via a joystick on a handheld game controller. Subjects are encouraged to travel to the target chest as quickly as possible and receive bonus points for efficient navigation.

and in the Results, on page 5:

These navigation periods represent epochs when the subject was fully in control of their movement in the environment, with the goal of quickly reaching the target treasure chest.

The reviewer had asked about the possibility of directly comparing moving versus stillness within navigation. We have examined this issue and, as we now explain in the manuscript, found that our task was not well suited for this comparison due to its fast-paced nature. The subjects were instructed during navigation to drive as quickly as possible to the currently visible treasure chest. Thus, as a result, subjects were moving forwards/backwards or turning for the vast majority of their time (96%) during "navigation" epochs. Furthermore, stillness epochs were very brief (median 320 ms) when they did occur. As a result of these factors, our task did not provide much data for examining neural correlates of stillness during "navigation." In the revised discussion (page 8), we now elaborate on this topic, explaining that we could not probe this issue due to our task design:

It is worth noting that our "navigation" condition, during which translational and rotational movement was under complete control of the subject, consisted of movement for 96% of all timepoints. As such, we did not have adequate data to examine neural signals related to pauses during navigation, where navigational planning may occur.

Finally, following the reviewer's suggestion, in the revised Discussion (page 8), we now acknowledge the possibility that the navigation-related activity we observed could in fact be a neural correlate of motor control. We explain that our task did not provide an control period for identifying this phenomenon and point out that this could be an interesting area for future work:

One critique of virtual reality studies of spatial cognition is that it can be challenging to be sure that a neural signal is specifically related to navigation rather than to any sensory or motor processing inherent to moving through virtual space [Taube et al., 2013]. Thus, it would be helpful for future iterations of virtual reality paradigms to include control conditions to distinguish signals related to sensorimotor processes. However, there is evidence that navigation-related hippocampal theta oscillations found during virtual tasks are not the result of low-level sensorimotor processes. For example, Vass et al. [2016] showed that movement-related theta oscillations persisted in the absence of motor control or optic flow when subjects were "teleported" across an environment while the screen was blank. This type of result, as well as others [Aghajan et al., 2016], suggest that our findings are relevant for real-world navigation.

- I am not convinced that the appropriate statistics have been used to establish the laterality of memory encoding and navigation effects. First, it is not clear to me whether recalled > not recalled and navigation > baseline are equivalent conditions, and therefore whether including them in the single ANOVA described on page 8 (with results shown in Figure 6) is valid. I am happy for the authors to keep this analysis in the manuscript, but it should be supported by additional statistics. In particular, the authors should include separate ANOVAs to assess the laterality of theta oscillations related to memory and navigation: i.e. one ANOVA with factors of hemisphere (left or right) and memory condition (recalled or not recalled); and one ANOVA with factors of hemisphere (left or right) and navigation condition (navigation or pre-trial baseline). In each case, laterality would be supported by a significant interaction between hemisphere and condition.

As suggested, we have performed these additional statistics and now them report them in the Results section alongside the original analysis on page 6. The outcomes of these new statistics are all consistent with the statistics we reported previously. The new text describing these analyses is as follows:

We confirmed these findings using a slightly different approach by computing individual two-way ANOVAs for the memory and navigation effects. Here, the factors were *hemisphere* and either *memory-success* or *navigation-state*. In this framework, the interaction term from the ANOVA can be interpreted as the strength of the lateralization of either memory- or navigation-related activity. The results of these tests mirrored the findings described above. The *hemisphere* × *memory-success* ANOVA resulted in a significant two-way interaction ($F(1, 45) = 5.97, p < 0.05$), replicating the above-described t-test showing a lateralized low-theta memory effect between the left and right hemispheres. Likewise, the ANOVA with factors of *hemisphere* and *navigation-state* again replicated the results of above-described t-test, showing a trend for an interaction ($F(1, 45) = 3.02, p < 0.1$).

- Second, to further support these results, the authors should assess whether power values are different from zero in each condition and hemisphere - i.e. it appears from Figure 6 that low theta power for recalled > not recalled is only greater than zero in the left hemisphere, but theta power for navigation v baseline is significantly greater than zero in both hemispheres. If this is the case, then the authors cannot claim that navigation is associated with increased low theta power in the right hippocampus only, rather that the magnitude of low theta power increases during navigation are (marginally) larger in the right hippocampus. This is a subtle but important distinction.

We apologize if our manuscript had been unclear on this point, but our main finding concerning navigation is that there is a statistically significant increase in theta power from zero in the right hemisphere and not in the left hemisphere. There was a trend indicating that the size of this increase was perhaps larger on the right than on the left ($p = 0.089$).

We have revised the text to be clearer regarding this issue by more precisely describing the conditions that cause statistically reliable increases in the power of theta oscillations in particular regions, and we also now include significance indicators for these tests directly in Figure 6. In the Results on page 6, we now state that:

Post-hoc tests for the navigation condition showed low-theta power was significantly greater during navigation than baseline in the right hemisphere only (right: $t(20) = 3.65, p < 0.01$, left: $t(20) = 1.43, p > 0.1$). A direct comparison revealed that there was a trend for the navigation effect to be greater in the right hemisphere than in the left (two-sample t-test: $t(45) = 1.73, p = 0.089$). For the memory condition, low-theta power was significantly greater for remembered items than forgotten items in the left hemisphere only (left: $t(25) = 3.72, p < 0.01$, right: $t(25) = 0.01, p > 0.1$). A direct comparison revealed that the effect was a significantly greater in the left than in the right (two-sample t-test: $t(45) = 2.44, p < 0.05$).

- Third, it is not clear to me why the authors show average t-statistics in Figures 3 and 6, nor is this correct. These are group level analyses, and therefore a single group level t-test should be performed on average z-scored power values (or average difference in power between conditions) across patients. The authors must not perform statistics on statistics and incorporate differences in within-subject variance into between-subjects analyses. Hence, they must change Figures 3 and 6 to show average power changes across subjects, and report the outcome of a single t-test on those power values across subjects at each cortical location / for each contrast.

Following this suggestion, we have recomputed all analyses using the difference in mean z-scored power instead of t-statistics as our measure of electrophysiological effects. None of the group-level results differ in their statistical outcome as a result of this change.

Across the entire manuscript, every result that was previously calculated based on subject-level t-statistics is now computed based on subject-level changes in z-scored power.

- Fourth, I am assuming that the ANOVA described on page 8 (with results shown in Figure 6) treats data from left and right hippocampi as independent samples, given that they can be drawn from different patients. However, claims of laterality might be further supported by repeated measures comparisons of data from each hemisphere in the same patient, where patients with bilateral depth electrode implants are available. This analysis should also be included.

The reviewer is correct that the referenced ANOVA treats data from left and right hippocampi as independent samples. Unfortunately, as mentioned above, patients with bilateral hippocampal electrodes are rare in our dataset and only account for 10 patients in total. Nonetheless, we have performed the suggested repeated-measures analysis and the findings largely mirror the main lateralization effects we reported previously. The results of the repeated-measures analysis are as follows: In the low theta band, there is a trend for an interaction between hemisphere and condition ($F(1,9) = 3.3, p = 0.1$). There is a trend towards a reliable memory effect in the left hippocampus ($t(9) = 1.9, p < 0.1$) but not the right ($t(9) = 0.6, p > 0.5$). Conversely, as expected, there is a significant navigation effect in the right hippocampus ($t(9) = 3.8, p < 0.01$) but no such effect in the left ($t(9) = 0.9, p > 0.3$).

The analysis of this dataset from the patients with bilateral coverage is shown above in Response Figure 2 and is also included in the paper as Supplementary Figure S5C. This information is described in the Results section on page 6 and statistical details are provided in the caption to Supplementary Figure S5C.

- Fifth, it would be useful to mention, at least briefly, the results of similar analyses on the 3-10Hz theta and 40-100Hz gamma bands, to further ascertain the specificity of the reported effects to the low theta band. In fact, data from all three frequency bands could be included as an additional factor in each of the ANOVAs described above, to establish whether there are any main effects of, or interactions with, frequency band.

As suggested, we performed similar analyses on other frequency bands in addition to our focus on low theta. These other bands do not show significant effects, which we think substantiates our decision to focus on the low theta band. We illustrate this point by now describing in the Results section of the main manuscript (Page 6) the outcome of two-way ANOVAs in the 3–10-Hz and 40–100-Hz bands with factors hemisphere and memory (good or bad) and with factors hemisphere and navigation/baseline:

Using this ANOVA framework, we also tested each of these effects in the 3–10-Hz and 40–100-Hz bands. We did not find any significant interactions in any of these four additional tests (all p 's > 0.3), indicating that there was no lateralized memory- or navigation-related hippocampal activity outside of the low-theta band.

We believe that this improves the manuscript by justifying our decision to focus on low-theta activity in the manuscript.

- Finally, the authors state that "...navigation-related low-theta power was greater in the right hemisphere than the left ($t(45) = 1.91, p = 0.06$)", but their statistical analysis clearly does not support this conclusion. This sentence must therefore be moderated to acknowledge that this is a weak and non-significant effect.

We have carefully edited the complete text of our revised manuscript to ensure that we describe any effects as non-significant trends if the associated p value is not below 0.05. The revised section of this text now describes this result as follows:

A direct comparison revealed that there was a trend for the navigation effect to be greater in the right hemisphere than in the left (two-sample t -test: $t(45) = 1.73, p = 0.089$)

- It is not clear to me why the authors lump both theta bands together in their decoding analyses, given that the vast majority of significant results mentioned above are specific to the low theta (1-3Hz) band. If successful discrimination cannot be supported by power changes in the low theta band only, then this should be mentioned as an important caveat to the preceding findings. Why not perform this analysis with each frequency band separately?

In hindsight we can understand why the reviewer was confused about this issue and we have edited the manuscript to more clearly articulate our rationale for this analysis. We now explain that the motivation behind our decoding analyses was to integrate brain-wide activity, which includes not only the low-theta signals that were prevalent in the hippocampus but also the faster neocortical signals. For this reason, we used the broader 1–10-Hz band as our general measure of low-frequency activity because this band would capture memory-related activity for classification in both hippocampal and neocortical regions (as seen in Figures 3 & 4A).

This issue is now explained on Page 7 as follows:

We used the wider 1–10-Hz low frequency range as a general measure of low-frequency activity in order to account for our finding that the hippocampus and neocortex exhibited memory-related signals at somewhat differing frequencies within this larger range (see Fig. 4A).

- **Abstract:** “Brain signals” seems a little simplistic for a scientific journal. I suggest “local field potential”, “EEG activity” or similar

As suggested, we changed this term to “intracranial electroencephalographic activity.”

- **Introduction:** “The theta oscillation in rodents is specifically hypothesized to be involved in memory because its amplitude correlates with memory encoding [Lega et al., 2012, Seager et al., 2002]” - neither of these citations describe rodent studies. The former examines theta activity in humans, the latter in rabbits. Hence, this sentence is misleading. For studies linking rodent theta activity with memory performance, I suggest Winson (1978) and / or McNaughton et al. (2006): Winson J (1978) Loss of hippocampal theta rhythm results in spatial memory deficit in the rat. *Science* 201: 160-163 McNaughton N, Ruan M, Woodnorth M-A (2006). Restoring theta-like rhythmicity in rats restores initial learning in the Morris water maze. *Hippocampus* 16: 1102-1110

We thank the reviewer for pointing out this mistake, and we have adjusted the citations as suggested.

- **Introduction:** “The statistical power of our study is greater than earlier work of this type [Miller et al., 2013, Suthana et al., 2012, Jacobs et al., 2016] because we enrolled 46 subjects and because each task session included 100 memory encoding events” - Again, I find this sentence a little odd, as none of the cited studies assess theta activity during memory encoding or spatial navigation, so I am not convinced that they constitute “earlier work of this type”. Presumably, a better comparison would be made with earlier human intracranial studies of theta activity during navigation, which generally have smaller sample sizes. However, I also note that several earlier human intracranial studies of theta activity during memory function use similar or much larger sample sizes (i.e. Lega et al. 2011, Burke et al. 2014).

We have edited the text on Page 3 to be more precise that we intended to compare our experiment with previous studies of the electrophysiology of successful versus unsuccessful spatial memory encoding. The revised text reads is as follows:

Our study is the largest investigation of the direct electrophysiological correlates of human spatial memory encoding. In particular, our experiment is notable because each session includes 100 spatial-memory encoding events, which provides substantially more data per session compared to earlier spatial memory paradigms (Miller et al., 2013, Suthana et al., 2012, Jacobs et al., 2016).

- **Methods:** “46 patients (28 male, mean age: 31.1 years) patients” - ‘patients’ is repeated

We have corrected this typo.

- **Methods:** “Here, subjects are shown a randomly selected object from that trial (Figure 1A, inset 4) and asked to move a target circle to indicate the location in the environment where it had been encountered. Subjects indicate their memory confidence on a three-level scale (Yes/Maybe/No), and then, from the same vantage point, use the joystick to move a target circle (radius 13 virtual units) to where they think the item was located (Figure 1A, inset 5)” - I am confused, are the participants asked to indicate the remembered location of an object before or after the confidence judgement, or both?

We have now modified the text of the Methods to more clearly describe the structure of our task. We now state (Page 11) that participants are first asked to rate their confidence and then to indicate the remembered location.

Here subjects are shown a randomly selected object from that trial (Figure 1A, inset 4), and on-screen text asks “Do you remember where to find the <object>?”. Subjects then select their memory confidence from the choices “Yes”, “Maybe”, or “No”. After this confidence response, subjects use the joystick to control a target circle (radius 13 virtual units) that is placed on the beach, and they indicate the location in the environment where they believe the object had been encountered.

- **Methods:** “Power values at each frequency were z-transformed within session” - to be clear, was the z-scoring performed across all epochs (i.e. pre-trial, memory encoding, and navigation) on the same electrode within each session? Please provide some more details here.

As requested, we now provide additional details on how the data was normalized for our statistical analyses. The “Spectral Analyses” section of the revised Methods section now explicitly states that the z-scoring was performed across all epochs within a session, separately for each frequency and electrode. The revised text reads as follows:

The resulting power values were then log-transformed (with the exception of the individual electrode power spectra shown in Figure 4B, we z-transformed the log-power values within session). This z transformation was performed separately for every session, electrode, and frequency, by subtracting the mean log-power across all event types (encoding, navigation, and baseline) and dividing by the standard deviation.

For time-frequency spectrograms (Figure 4), we averaged log-power into 69 overlapping windows of 100 ms each, in steps of 50 ms, between -1500 and 2000 ms relative to item onset. We then z-scored as described above, now separately for session, electrode, frequency, and timepoint. For the longer time periods shown in Figure 5, we averaged log-power into 56 overlapping windows of 500 ms each, in steps of 100 ms, between -2250 and 3750 ms relative to item onset, and we normalized the z-power values based on the mean power of the baseline condition.

- **“Euclidean” should be capitalized.”**

We have corrected this typo.

Reviewer 2

- **Despite authors’ claims, the treasure hunt task does not build on the classic water maze and in fact, is different from that task. Specifically, the utilized paradigm is a guided navigation task and subjects move towards visible targets, as opposed to finding hidden locations in the water maze. Spatial learning occurs in the form of object-place associations, only when subjects have reached the treasure chests, and not during movement.**

The reviewer makes a good point, and we agree that the manuscript could have been more precise in describing the distinctive features of the task. We have edited the relevant section of the Introduction accordingly and, in particular, no longer state that our methods build on Morris's water maze:

TH draws inspiration from both the human verbal memory and rodent spatial navigation domains and asks subjects to memorize multiple object locations while virtually navigating an open arena. As such, TH can be thought of as a spatial paired-associate learning task in which participants memorize object–location pairs—this type of task is known to be hippocampally dependent [Schwarb et al., 2016]. TH's design includes separate time intervals for memory encoding and navigation, allowing us to distinguish the neural correlates of these processes.

- Indeed, it is a peculiar choice to switch between first-person navigation to a location for the encoding and simply marking the correct location from afar. In the least, some additional analyses are in order, as well as more detailed description of the task. Are items that are located farther from the point of view during testing less accurate than nearby items?

We thank the reviewer for these comments and we agree that it will improve the paper to elaborate on the task design and its rationale. In the revised Methods (Page 11), we explain that we chose to have a third-person view during retrieval to save time, which we felt was advantageous given our focus on maximizing the number of memory encoding events per session:

The reason for this perspective shift was to speed the retrieval period, conserving patient testing time to provide a relatively larger number of memory encoding events. Additionally, the retrieval location randomly alternated between two ends of the environment to encourage the use of an allocentric strategy.

The reviewer asked whether items that were located further from the point of view during testing were remembered less accurately than closer items. We examined this issue by separately computing mean accuracy for near and far memory items, as determined by splitting the environment in half based on distance from the testing location. There was slightly higher memory accuracy for near items than far items (0.71 and 0.67, respectively; $t(49) = 4.9, p < 0.001$). Given this finding, we thus examined whether the low-theta subsequent memory effect we observed in the full dataset differed as a function of near vs far items. Focusing on the left hippocampus, we computed the memory-related change in low-theta activity separately for near and far conditions. We found no significant difference between the two. Critically, this result indicates that the memory-related theta activity reported in the manuscript was separate from the item position effects. We now describe this analysis as in the revised manuscript as follows (Page 4):

Behaviorally, subjects were more accurate in locating items encountered in the near half of the field relative to the testing location compared to the far half (0.71 vs 0.67; $t(49) = 4.9, p < 0.001$). However, this behavioral difference did not manifest as differences in neural activity, as the magnitude of the left-hippocampal low-theta memory effect was similar for both near and far items ($p > 0.1$).

- Do subjects see the first treasure chest from the testing vantage point but then subsequent treasure chests only from within the beach? This could cause the first item to be more accurately placed due to its greater similarity between training and testing, rather than any memory effect. Were subjects ever tested on the location of empty chests; were they informed that they would not have to remember the locations of those items?

We have revised the description of the task to clarify the questions that were raised here. We now note that all of the chests are only learned during the encoding period when the subject is on the ground and that subjects were never tested on the location of empty chests (because they contained no item to probe). The relevant revised section of text reads as follows (see Page 10):

On each trial, the subject is placed on the ground at the edge of the environment, in either the hut or the semicircle of totem poles ... after traveling to the fourth chest, the subject is smoothly moved to one of the ends of the beach where they have an elevated 3/4 overhead perspective view of the environment.

- As stated above a critical issue pertains to lateralization. When considering lateralization of function as studied in epilepsy patients, it's important to consider the lateralization of seizure onset. I'd like to see a supplementary figure showing that the results hold up, even when limiting analysis to electrodes that are fully outside of the seizure onset zone. The methods should also describe how interictal baseline activity was treated when analyzing the physiology.

These important points were also raised by the editor and by Reviewer 1, and they are discussed in more detail in the opening section of our response. In short, we reformed our analyses while excluding a subject's hippocampus if any medial temporal lobe electrodes in the same hemisphere fell in a seizure onset zone. Performing this exclusion did not yield in meaningful changes to the pattern of results, indicating that our findings are not due to epileptic pathology.

Additionally, as suggested, we have revamped our analyses to exclude experimental events using a kurtosis threshold of 5 to identify potential epileptiform activity (see also response to Reviewer 1). This did not yield any substantial changes to our results.

- In their localization figure, Figure 2B, it seems that their automatic segmentation algorithm has not worked well. It appears that there are areas shown as subiculum where there supposed to be Dentate gyrus, CA2 and CA3 areas.

After consultation with our neuroradiologists, we believe the segmentation is correct. CA2 and CA3 are not present in the slice shown given that the location is in the very anterior portion of the hippocampus. However, we have modified the color mappings used in Figure 2 to more clearly demarcate the individual subregions.

To give confidence show that our algorithms are indeed working correctly, Response Figure 6 illustrates the results of our segmentation algorithm positions along the hippocampus for this same patient. The left panel shows the same slice as shown in the original Figure 2. Here, CA1 (red), dentate gyrus (purple), and subiculum (pink) are all visible, whereas CA2 (green) and CA3 (yellow) are not. Moving 20 slices more posterior (0.6 cm) reveals CA2 and CA3.

Response Figure 6: Two coronal slices with medial temporal lobe subregions color-coded based on our automatic segmentation procedure. CA1: red, CA2: green, CA3: yellow, subiculum: pink, dentate gyrus: purple, entorhinal cortex: tan, BA 35: light blue, BA 36: dark blue. Voxel spacing along z-axis: 0.3 mm.

- Please report Ns and other relevant statistics in your figure legends.

We now report Ns and define statistical results in all relevant figures.

- Additionally, why is using t-statistic justified? Are the distributions normal?

The use of t statistics is appropriate for our comparisons because we had already applied a logarithmic transformation to the raw power values, which are log-normal. Therefore, the measures that enter our t test are normally distributed.

- In figure 6 bar plots, it would be nice to see individual data points as well as box plots.

We agree that individual data points convey useful information. Supplementary Figure S5 replicates Figure 6 for the full dataset, the dataset excluding seizure onset zones, and the bilateral only subjects—these plots all include individual subject data points.

- Statistics in Figure 4: It is not statistically appropriate to calculate the mean difference of z-scored distributions (in this case power). The authors might have to think about a better way to normalize their data as opposed to z-scoring.

We feel there may be a misunderstanding here, because our methods for analyzing z-normalized data are extremely common when performing spectral analysis of neural data [e.g. Sederberg et al., 2006, Burke et al., 2014, Long et al., 2014, Yaffe et al., 2014]. Further, our approach is a recommended method in Michael X Cohen's textbook "Analyzing Neural Time Series Data".

Perhaps there is some confusion with exactly how the z-scoring is done. If, for example, we z-scored the remembered and forgotten items separately, then it would not make sense to compare the distributions as both would have a mean of 0 and standard deviation of 1 (and thus there would be no statistical difference between them). However, that is not the method we utilized. Instead, we performed z-scoring across all experimental events within a session, regardless of remembered or forgotten condition (or navigation or baseline). We have modified our methods section, as also suggested by Reviewer 1, to ensure that our description of the z-scoring method is clear (see Page 12 or the earlier section of this letter).

- Also, please include raw traces from several trials showing the oscillations.

As requested, we now include example theta oscillations measured during memory encoding from hippocampal contacts from several trials in the new Supplementary Figure S2.

- Using a median-split within patient to define remembered vs forgotten doesn't seem fair, as some patients likely remembered most items and others might well have forgotten most. Perhaps an across-subject split would be better. If the across patient accuracy distribution were bimodal and you could split between the peaks, that would be ideal. In either case, please show the distribution and the location of the split.

It is certainly true that some subjects are better than others (the distribution of mean accuracy is shown in Figure 1 of the main text). This intersubject variation was, in fact, part of our rationale for using a within-patient median split. Doing so ensured that all patients had both remembered and forgotten items, regardless of their absolute level of performance. An across-subject split could result in some subjects having very few (or no) items in one condition, which could bias the results. Given these factors, we felt that performing a within-patient split was the best solution, especially because this method helps account for interindividual differences in performance that may be due to factors that are not of interest, such as inexperience with video games or game controllers.

Nonetheless, we performed a follow-up analysis to ensure that our particular results were not influenced by our use of this median split. Both our independent variable (accuracy) and our dependent variable (z-scored power) are continuous values and, thus, when analyzing memory-related changes in neural activity, we can avoid the discretization of items into remembered and forgotten. Instead of computing a change in z-scored power between the memory conditions, we computed a correlation coefficient that measures the relationship between accuracy and power at every electrode and frequency. To compute group-level statistics, here we use a t-test to compare the distribution of correlation coefficients (instead of changes in power) for a given region and frequency band of interest to an expected mean of zero. Response Figure 7 shows a comparison of the hippocampal memory effects using the original median-split method (left panel) and the correlation method (right). As this plot shows, the outcome of this new analysis is quite similar to the results of our median-split method, indicating that the method used in the manuscript does indeed capture subject performance well. For the purposes of facilitating comparisons with navigation and baseline conditions, we focus on the results of the median-split method in the revised manuscript. However, we would be willing to include this analysis if the reviewers or editor felt it would be helpful.

Response Figure 7: Subsequent memory effects for hippocampal electrodes. Left: Distributions comprised of changes in z-scored power for remembered minus forgotten items. Right: Distributions comprised of Pearson product-moment correlation coefficient between accuracy and z-scored power.

- **Classification:** it seems like the main claim of the paper is that the differential contribution of the oscillatory activity to successful recall and navigation occurs in the low theta band. However, in the classification algorithm, the input consists of the 1-10Hz frequency range. If you used low-theta range of 1-3Hz, is the classification algorithm still successful?

As mentioned in response to a similar comment by Reviewer 1, we elected to use the 1–10 Hz range as a general measure of low-frequency activity due to the fact that the neocortical memory effects we observed spanned this whole low-frequency range and, in fact, appear to be stronger in the high-theta band. We now explain this issue directly in the results, on Page 7:

We used the wider 1–10-Hz low frequency range as a general measure of low-frequency activity in order to account for our finding that the hippocampus and neocortex exhibited memory-related signals at somewhat differing frequencies within this larger range (see Fig. 4A).

- In the same way that the authors determined the significance of t-stats using the 2.5-97.5%ile of shuffled values, it would be instructive to see the same thing for the AUC analysis instead of a t-test compared to 0.5. Critically, the authors should shuffle the remembered/forgotten labels of the trials without changing the total number to receive each label.

As suggested, we have now also used a non-parametric test to evaluate the statistical significance of our classification results. We now compute a null distribution of AUC values based on 100 permutations of each subject's remembered/forgotten labels, and we compare the true mean AUC value to the 100 means derived from shuffled data. For both the low- and high-frequency ranges, the true mean AUC was greater than every shuffled AUC (permutation p 's < 0.01). This provides strong evidence that our classification is reliably above chance. We now report these statistics alongside the parametric tests, on Page 7:

We also assessed significance using a shuffling procedure, whereby we computed a null distribution of AUC values based on 100 permutations of each subject's good memory and bad memory labels, and we compared the true mean AUC value to the 100 means derived from shuffled data. For both the low-frequency and high-frequency ranges, the true mean AUC was greater than every shuffled AUC (permutation p 's < 0.01).

- Page 5, line 93: "Overall, power in the low-theta and theta bands was significantly greater for recalled items compared to forgotten ones." At least according to Figure 3 (middle row), this statement is not true for theta band, as the values do not pass significance levels.

We now have clarified this introduction to our results. We now state that the overall trend across our entire dataset is the presence of increased activity at low frequencies for good memory; following sections then more precisely indicate the set of brain regions that show individual effects in more detail. The revised text (Page 4) reads as follows:

Overall, the general trend across our whole dataset was that power in the low-theta and theta bands was elevated for recalled compared to forgotten items. In the neocortex, this theta effect was most prominent for electrodes in the anterior lateral temporal lobe, with weaker effects in the HFA band or in other surface regions (Figure 3A). We next turned our attention to depth electrodes, given our interest in the role of medial temporal lobe structures in spatial memory. We found a 1–3-Hz power increase for recalled items in the left hippocampus ($t(25) = 3.72, p < 0.01$).

- Figure 5: In figure 4, the authors show that in the left hippocampus the difference in power for remembered vs forgotten trials begins half a second before the chest opens. In figure 5, however, the time course for power in the no-item case very closely follows the not-subsequently-remembered trace (in both low-theta and HFA). This is suspicious, as the subjects should not know ahead of time whether a chest will have an item in it, and warrants further exploration/discussion.

We understand the reviewer's comment, however, as we now clarify in the methods (Page 10), subjects cannot predict whether a chest will be empty or filled because we randomized the number of filled chests (either 2 or 3) in each trial. Unlike the comparison between the subsequently-remembered and not-subsequently-remembered low-theta timecourses, which show statistically significant differences beginning 600 ms before the item appears, there are no statistically reliable pre-stimulus differences between no-item timecourse and either of the memory conditions in any of the three frequency bands in either hemisphere. Thus, the visual similarity between conditions noted by the Reviewer is unlikely to reflect a true difference between conditions.

Because the number of chests containing an object randomly varied between two and three, subjects could not predict whether the current target chest contained an object. This served to remove effects of expectation and to encourage subjects to always attend to their current location as they approached a chest.

- The description of how “accuracy” was calculated is inadequate. Please include an equation or code, such that the reader could reproduce the same measurement. It is unclear that 0.5 is chance. Perhaps the authors could add a supplementary figure that shows several different overhead maps of the beach with different target locations, and color code the beach by the accuracy that would be assigned if the subject had selected that location.

We have created a new supplemental figure to better explain how our accuracy measure is calculated. This figure is shown below as Response Figure 8 and is also in the revised paper as Supplementary Figure S1.

In this figure we demonstrate the accuracy calculation for an object in the corner and an object in the center of the environment. Panel A shows all possible Euclidean distance errors for these two locations, where the possible maximum error is greater for the object located in the corner. Panel B shows how a single error distance of 20 virtual units corresponds to a different percentile in the distributions of possible errors according to the object location. Given the possible response locations, an error of 20 units for the corner object corresponds to better memory performance than an error of 20 units for the center object, as random guessing could result in a lower error for the center object simply due to the distribution of possible errors. In this measure, the reported accuracy is 1 minus the percentile of the error. A useful feature of this method is that it is unbiased in the case of random guessing, whereas the raw Euclidean distance metric is not, because it is affected by the shape and range of the distribution of possible errors for each response location.

Response Figure 8: **A.** Overhead heat maps showing the possible Euclidean distance errors for an object located in the corner (left) and the center (right) of the environment. **B.** Probability distributions of possible Euclidean errors for the locations shown in A. The vertical line shows where a response of 20 units falls relative to all possible responses.

- When describing the use of bipolar electrodes, please specify the electrode spacing. Was the location of the "electrode" taken as the midpoint between the two?

This information has been added to text, on Page 11:

The location of these new virtual electrodes was taken to be the midpoint between the two physical contacts (inter-electrode spacing: 10 mm). Data from the virtual electrodes were used in all analyses and are referred to as in the text as “electrodes”.

- Supplementary Figure 1: Are all of those 10 neurons recorded from the left hippocampus? If not, how does this support the main claim?

We have revised this section of text to describe our intent more clearly. The primary point we tried to make with the inclusion of the supplemental figure was not to focus on lateralization but instead to show the general correspondence between HFA and single-neuron spiking during a memory task, which has not been reported in any form to our knowledge in humans.

To explain this point, we have clarified the text on Page 5 to explain that while the effect is strongest in the left hemisphere, Figure 5 indicates that the right hemisphere also appears to show greater HFA activity for filled chests. We think that this small change is important because it underscores this general point that both HFA and neuronal spiking increased for viewing a filled chest in both hemispheres. Unfortunately, we did not have sufficient single-neuron recordings from both hemispheres to analyze the potential lateralization of this phenomenon in detail.

Reviewer 3

- While navigation epochs do involve navigation, they don't test the type of navigation explored in most studies on the hippocampus - allocentric based navigation to a hidden goal. Rather the authors are tapping into vista-space approach - see e.g. Wiener, J. M., Buchner, S. J., & Holscher, C. (2009). Taxonomy of human wayfinding tasks: A knowledge-based approach. *Spatial Cognition & Computation*, 9(2), 152-165. Moreover its not clear they are tapping into navigation at all with their contrast to the baseline of not moving, such that it would appear that R-hippocampal 1-3Hz may be driven by visual motion. Given this I think the authors need to be more careful in their framing about what they are referring to as navigation and discuss this issue in more detail.

We agree that the navigation phase of our task has some features reminiscent of vista-space processing. However, for several reasons, we believe the overall task design requires the use of an allocentric representation to support spatial memory. We disorient the subjects between encoding and retrieval (when they are moved to the retrieval location). This means that in order for them to respond accurately, they must orient themselves and then recall the location of the cued object in the global allocentric frame, rather than by following a fixed series of movements during response. In addition, because we probe each item individually, out of order, subjects cannot perform the task accurately by simply recreating their egocentric route. While we agree that the act of driving to a marked chest can be accomplished with a vista-based strategy, the subjects would need to quickly reinstate their allocentric representation for memory encoding upon chest arrival. Therefore, although it is impossible to infer precisely what proportions of time the subject is using vista or allocentric strategies, we can definitively state that allocentric representations were frequently and accurately accessed during the task.

The reviewer points out that some apparent navigation-related activity could relate to visual motion. This reflects a general weakness in using virtual reality for studying navigation because subjects only perceive moving through an environment via optic flow rather than via proprioception. However, a recent paper by Vass et al. [2016] showed evidence that human hippocampal theta correlated with translational movement even when subjects did not receive optic flow in a "teleportation" condition. Although Vass et al. did not analyze the lateralization of their effects, we now cite and discuss this work, as it indicates that findings of human hippocampal theta are not purely the result of visual motion.

As was mentioned above in response to Reviewer 1's comment about confounds due to motor control, and as we now state in the revised Discussion:

One critique of virtual reality studies of spatial cognition is that it can be challenging to be sure that a neural signal is specifically related to navigation rather than to any sensory or motor processing inherent to moving through virtual space [Taube et al., 2013]. Thus, it would be helpful for future iterations of virtual reality paradigms to include control conditions to distinguish signals related to sensorimotor processes. However, there is evidence that navigation-related hippocampal theta oscillations found during virtual tasks are not the result of low-level sensorimotor processes. For example, Vass et al. [2016] showed that movement-related theta oscillations persisted in the absence of motor control or optic flow when subjects were "teleported" across an environment while the screen was blank. This type of result, as well as others [Aghajani et al., 2016], suggest that our findings are relevant for real-world navigation.

- More detail on the subsequent memory effects should be included. It is possible that a certain set of objects are commonly remembered and thus visual stimuli effects are driving the subsequent memory effect rather than a true subsequent memory effect.

This comment and the following one both refer to the interesting possibility that our analyses could lead us to call something a memory effect when it is actually more directly related to a different aspect of the task. We examined this issue and we now report that memory-related theta activity seems to occur independently of stimulus features and location.

We now mention this analysis in the revised Methods section when describing the task, on Page 11:

To rule out the possibility that our neural analyses of subsequent memory could be influenced by variability in the memorability of the objects in our pool, we analyzed mean recall performance as a function of object identity. We performed a one-way repeated-measures ANOVA with object identity as a within-subject factor to determine if there was a reliable difference in performance from item to item. We did not find significant effect of object identity ($F(100, 4900) = 1.09, p > .2$), leading us to conclude that variability in the recall rates for individual objects was not strong enough to drive the observed neural memory effect.

- Similarly, it would be useful to know whether participants were more accurate for items towards the corners of the environment - and if so does this relate to the subsequent memory effect observed?

Based on our recent work with similar tasks (Lee et al., in revision; Goyal et al., in revision), we would indeed expect higher performance for objects located near the boundaries of the environment compared to objects near the center. We examined this issue in our dataset by dividing the environment into an “inner” region and a “boundary” region, where the inner region is a rectangle with the same aspect ratio as the whole environment and comprised of half of the total area, and the “boundary” region is the remaining outer area. We found that mean memory accuracy was lower for the inner region compared to the boundary region (0.66 and 0.71, respectively; $t(49) = 5.2, p < 10^{-6}$).

Next, we tested if these boundary-related performance differences explained the subsequent memory effect that we observed. To do this, we separately computed the memory-related change in low-theta activity in the left hippocampus separately for both regions of the environment. The magnitude of this subsequent memory effect was similar in both inner and boundary regions (t -test $p > .2$), indicating the overall memory-related pattern remained stable across the environment despite the changes in the absolute recall rates between regions. This analysis is explained in the revised text on Page 4:

... Similarly, subjects exhibited more accurate memory performance for items studied near the boundaries of the environment compared to items studied closer to the center (0.71 vs 0.66, $t(49) = 5.2, p < 10^{-6}$, see *Methods*). However, this boundary-related performance boost did not seem related to the left hippocampal low-theta memory effect because the magnitude of this effect was similar when separately calculated for items both near and far from boundaries ($p > 0.2$).

- It wasn't clear how often the empty chest occurred - it would be useful when this is introduced to say this frequency. This is important when interpreting the neural responses - i.e. are these novel events?

Empty chests were not especially rare. Either 1 or 2 chests per trial were empty (37.5% of all chests in a session). This information is now included in Methods, on Page 11:

Subjects encounter 160 chests in a full session, 100 of which contain objects and 60 of which are empty (either one or two empty chests per trial).

- Other articles the authors may wish to cite that provide support their left-sided focus in memory retrieval and specifically object-location process are respectively: Maguire, E. A., & Frith, C. D. (2003). Lateral asymmetry in the hippocampal response to the remoteness of autobiographical memories. *Journal of Neuroscience*, 23(12), 5302-5307, Howard, L. R., Kumaran, D., ÅşlafsdÅştir, H. F., & Spiers, H. J. (2011). Double dissociation between hippocampal and parahippocampal responses to object-background context and scene novelty. *Journal of Neuroscience*, 31(14), 5253-5261.

We appreciate this helpful suggestion and have added these citations to our Introduction (see Page 2).

Reviewer 4

- To what extent is the lateralisation differences a result of stimulus type? The encoding task requires verbal processing whereas the navigation task is more visually focused. The second figure in Davachi (2006, *Current Opinion in Neurobiology*) demonstrates a bias in left hippocampal activation for words and right hippocampal activation for objects/faces/scenes. Could the increased left hippocampal theta power for the encoding task reflect the dominance of verbal material, and the increased right hippocampal theta power for navigation reflect the dominance of visual material?

In the revised manuscript we now discuss our results in the context of the point described by the Davachi review paper. Please see the “Response to comments from the editor” section above.

- Are there sufficient participants with recordings from both the left and the right hippocampus to conduct this analysis within-subjects? A common criticism of lateralisation differences between subjects relates to inter-group differences, but a within-subjects analysis would side-step this issue and would strengthen the claim of a lateralised hippocampus.

As suggested, have now performed a within-subject analysis and report these results in a new Supplementary Figure S5C. In brief, the new within-subject analysis largely replicates our earlier findings, indicating that our findings do not reflect interindividual differences. For detail, please see the "Response to comments from the editor" section and the response to Reviewer 1 on a related point.

- In figure 4, the differences in oscillatory power do not appear to be event-related, unlike a number of other subsequent memory effects (SME; e.g. Crespo-Garcia et al., 2016, *NeuroImage*; Staudigl and Hanslmayr, 2013, *Current Biology*). Do the authors envision that these findings are analogous to previous findings, or rather a distinct aspect of memory formation? Is the extended SME possibly related to the uniqueness of the task?

We agree that it is interesting that the memory effects in our task seem to be more spread over time compared with some earlier studies. In the revision on (Page 9) we now discuss possible explanations for this difference, as it could reflect a new physiological phenomenon or it could relate to the fact that subjects can see chests prior to arrival:

It is notable that the memory-related theta activity in our task was present up to one second before chest opening, in contrast to previous work that showed memory-related effects that followed stimulus onset (Burke et al., 2013; Crespo-Garcia et al., 2016; Staudigl and Hanslmayr, 2013). Future work will have to test whether this reflects the unconventional nature of our spatial task, in which subjects see each chest prior to arrival, or whether human memory-related theta oscillations simply appear on a wider timescale so that significant effects are apparent prestimulus, as seen in an earlier study (Guderian et al. 2009).

- A number of previous studies have reported broadband increases in high-frequency activity (HFA) relating to successful memory formation, together with low frequency decreases (e.g. Burke et al., 2014, NeuroImage; Long & Kahana, 2015, NeuroImage). In this report the authors report no significant effects in HFA with subsequent memory. Surprisingly they even report a tendency in the opposite direction. Together with the reported increased low frequency power increases which support episodic memory encoding there appears to be a possibility that the employed task somehow switched the time-frequency pattern associated with memory encoding, i.e. from low frequency power decreases/HFA increases low frequency power increases/HFA decreases. Can the authors offer an explanation for this apparent discrepancy?

We agree that it is interesting that the fundamental nature of the memory-related signals in our spatial task differ compared to earlier human verbal memory work (although our findings are similar to the earlier rodent literature). We think that this represents a novel feature of our human study, and we have modified our existing discussion of power increases/decreases to elaborate on this difference in the revised discussion. Here, we hypothesize that the nature of the SME pattern is different in our task due to its spatial nature (Page 9):

Theta power has been reported to both increase and decrease during item encoding (see Hanslmayr & Staudigl [2014] for an overview). In verbal memory tasks such as free recall or paired-associate learning of word lists, decreases are often reported [Burke et al., 2013, Long et al., 2014, Greenburg et al., 2015], but the direction of this effect may change depending on the exact time period analyzed [Guderian et al., 2009], or even, intriguingly, as a function of within-task manipulations of testing conditions [Staudigl & Hanslmayr, 2013]. Thus, in light of these earlier studies on the electrophysiology of human memory, it might be considered surprising that we found that the power of low-frequency signals positively correlated with memory encoding and, inversely, that the power of high-frequency activity (HFA) tended to correlate negatively. We hypothesize that these differences reflect the spatial demands in our task, such that low-frequency signals across the brain are more relevant during spatial processing, whereas memory tasks that are purely verbal rely on brain networks that correlate with HFA activations. Our findings on the role of low-frequency oscillations in spatial memory are consistent with earlier work in rodents showing that hippocampal theta power positively correlated with spatial memory encoding [Winson, 1978; McNaughton et al., 2006], and they build on previous noninvasive measurements from magnetoencephalography, which showed hippocampal theta positively correlated with performance in virtual navigation [Cornwell et al., 2008, Kaplan et al., 2012].

- Line 7. "...theta oscillations at 4-8Hz whenever this structure active..." - possibly missing an "is"?

We have corrected this typo.

- Line 67-67 and line 352. A normalised accuracy of 0.5 is referred to as chance here; however I feel that this assumes that the participant selects a random location on the map. If a location was unknown, I imagine that a sensible strategy to give a reasonably accurate response would be to select a location near the centre of the map, rather than at one of the corners, as this is perhaps the closest position to all possible locations. This would bias guessing (and chance performance). Perhaps the "chance" terminology should be dropped.

As suggested, we have removed the reference to "chance" level performance.

- Figure 3. Given the focus on the hippocampus, medial surface plots here would be of interest.

We agree that images showing the hippocampus directly would improve the manuscript. In the revised paper we have added new panels to Figures 2 and 3 that show medial slices through the hippocampus.

- Line 349. “we transformed the raw error distance for each response an “accuracy” score...”. A word or two may be missing from this sentence.

Thank you, we have corrected this typo.

References

- Z. M. Aghajan, P. Schuette, T. Fields, M. Tran, S. Siddiqui, N. Hasulak, T. K. Tcheng, D. Eliashiv, J. Stern, I. Fried, et al. Theta oscillations in the human medial temporal lobe during real world ambulatory movement. *Current Biology*, 27(24):3743–3751, 2016.
- J. F. Burke, N. M. Long, K. A. Zaghoul, A. D. Sharan, M. R. Sperling, and M. J. Kahana. Human intracranial high-frequency activity maps episodic memory formation in space and time. *NeuroImage*, 85:834–843, 2014.
- L. Davachi. Item, context and relational episodic encoding in humans. *Current Opinion in Neurobiology*, 16(6):693–700, 2006. doi: 10.1016/j.conb.2006.10.012.
- J. Jacobs. Hippocampal theta oscillations are slower in humans than in rodents: implications for models of spatial navigation and memory. *Philosophical Transactions of the Royal Society B: Biological Sciences*, 369(1635):20130304, 2014.
- N. M. Long, J. F. Burke, and M. J. Kahana. Subsequent memory effect in intracranial and scalp EEG. *NeuroImage*, 84:488–494, 2014. doi: 10.1016/j.neuroimage.2013.08.052.
- J. R. Manning, J. Jacobs, I. Fried, and M. J. Kahana. Broadband shifts in local field potential power spectra are correlated with single-neuron spiking in humans. *Journal of Neuroscience*, 29(43):13613–13620, 2009.
- H. Schwarb, C. L. Johnson, M. D. McGarry, and N. J. Cohen. Medial temporal lobe viscoelasticity and relational memory performance. *Neuroimage*, 132:534–541, 2016.
- P. B. Sederberg, L. V. Gauthier, V. Terushkin, J. F. Miller, J. A. Barnathan, and M. J. Kahana. Oscillatory correlates of the primacy effect in episodic memory. *NeuroImage*, 32(3):1422–1431, 2006. doi: 10.1016/j.neuroimage.2006.04.223.
- J. S. Taube, S. Valerio, and R. M. Yoder. Is navigation in virtual reality with fmri really navigation? *Journal of cognitive neuroscience*, 25(7):1008–1019, 2013.
- M. K. van Vugt, A. Schulze-Bonhage, B. Litt, A. Brandt, and M. J. Kahana. Hippocampal gamma oscillations increase with working memory load. *Journal of Neuroscience*, 30(7):2694–2699, 2010.
- L. K. Vass, M. S. Copara, M. Seyal, K. Shahlaie, S. T. Farias, P. Y. Shen, and A. D. Ekstrom. Oscillations go the distance: Low-frequency human hippocampal oscillations code spatial distance in the absence of sensory cues during teleportation. *Neuron*, 89(6):1180–1186, 2016.
- R. B. Yaffe, M. S. Kerr, S. Damera, S. V. Sarma, S. K. Inati, and K. A. Zaghoul. Reinstatement of distributed cortical oscillations occurs with precise spatiotemporal dynamics during successful memory retrieval. *Proceedings of the National Academy of Sciences*, 111(52):18727–8732, 2014.
- H. Zhang and J. Jacobs. Traveling theta waves in the human hippocampus. *The Journal of Neuroscience*, 35(36):12477–12487, 2015.

Reviewers' comments:

Reviewer #1 (Remarks to the Author):

I am satisfied that the authors have addressed all of my concerns regarding the previous version of this manuscript.

I have only one or two final minor comments.

- (1) "In a separate line of work, neuroimaging work" - it would be preferable to avoid repeating the word 'work', or in fact, using it at all. How about "In a separate line of inquiry, neuroimaging research"?
- (2) "TH's design design includes separate time intervals" - 'design' is repeated
- (3) "This pattern was also visible qualitatively in the right hippocampus" - but was the difference in high frequency power for item viewing v empty chests significant at any time point in the right hippocampus, or not? Please clarify
- (4) "This type of result, as well as others [Aghajan et al., 2016]" - it might be appropriate to cite the recent Bohbot et al. paper studying human intracranial EEG during real world navigation here too, but I am happy to leave that to the author's discretion
- (5) "Patient-specific hippocampal volumes were smoothed with a 4 mm gaussian filter before subsequent analysis." - 'Gaussian' should be capitalised

Reviewer #2 (Remarks to the Author):

The manuscript is improved and the authors have partially addressed the review concerns. The new colormap of localization does help. Remaining concerns:

- 1) They reanalyzed the data after excluding the electrodes in the seizure onset zones. However, they are not correcting their p-values for this multiple comparison. Since they are not reporting the exact p-values, I am not sure if their $p < 0.05$ will survive significance. This is regarding their old figure 6 and figure 1 in the rebuttal letter.
- 2) Regarding within subject comparisons of lateralization effect: If you look at their response figure 2, everything loses significance except the results from the right hippocampus showing more low-theta during navigation

Reviewer #3 (Remarks to the Author):

I have carefully reviewed the authors response and I am happy to endorse publication.

Reviewer #4 (Remarks to the Author):

I have read the revised version of the paper and the responses to my comments. I feel the authors have adequately addressed all concerns raised.

My only remaining concern is how the authors have presented the within-subjects analysis in the manuscript. I feel that the phrase "Despite the large reduction in the size of the dataset, we still observed the same general pattern of effects" is overselling the result a bit, given that some of the effects are not significant anymore (i.e. the interaction). This should be acknowledged and the statement toned down accordingly. Perhaps something along the lines of "while the interaction was no longer significant (quite possibly due to the large reduction in sample size), the general pattern of the effect was retained".

We thank the reviewers and editor for their additional comments on our manuscript. Our revised manuscript addresses each point raised by the reviewers. Below, red coloring indicates reviewer comments, and our responses are indicated below in black text.

Response to Reviewer 1

- “In a separate line of work, neuroimaging work” - it would be preferable to avoid repeating the word ‘work’, or in fact, using it at all. How about “In a separate line of inquiry, neuroimaging research”?

We have changed the text to match this suggestion.

- “TH’s design design includes separate time intervals” - ‘design’ is repeated

We have corrected this typo.

- “This pattern was also visible qualitatively in the right hippocampus” - but was the difference in high frequency power for item viewing v empty chests significant at any time point in the right hippocampus, or not? Please clarify.

The difference in high frequency power for item viewing compared to empty chests did not reach significance in the right hippocampus, though there was a strong trend in the time interval of 0.7–1.6 seconds relative to item onset ($p = 0.065$). We have added this clarification to the manuscript as follows:

This pattern was also visible qualitatively in the right hippocampus and trended towards significance in the 0.7–1.6-s time interval (cluster $p < 0.1$).

- “This type of result, as well as others [Aghajan et al., 2016]” - it might be appropriate to cite the recent Bohbot et al. paper studying human intracranial EEG during real world navigation here too, but I am happy to leave that to the author’s discretion.

As suggested, we have added a reference to the recent Bohbot et al. manuscript here, as we agree that it is relevant here.

- “Patient-specific hippocampal volumes were smoothed with a 4 mm gaussian filter before subsequent analysis” - ‘Gaussian’ should be capitalised.

We have corrected this typo.

Response to Reviewer 2

- They reanalyzed the data after excluding the electrodes in the seizure onset zones. However, they are not correcting their p-values for this multiple comparison. Since they are not reporting the exact p-values, I am not sure if their $p < 0.05$ will survive significance. This is regarding their old figure 6 and figure 1 in the rebuttal letter.

Regarding the analysis of the data outside of the seizure onset zone, we do not believe that a multiple-comparison correction is necessary here. This analysis is a single ANOVA that re-examines a subset of the data that was originally analyzed in order to confirm the previously described result. Nonetheless, to be responsive to the reviewer’s comment, we modified the text to now explicitly state the exact p value from this ANOVA, $p = 0.02$, which we think will ameliorate any potential concern concerning the robustness of this result.

- Regarding within subject comparisons of lateralization effect: If you look at their response figure 2, everything loses significance except the results from the right hippocampus showing more low-theta during navigation

A similar concern was also raised by Reviewer 4, and we agree that we perhaps overstated the strength of the within-subject results. Our primary goal was to convey the idea that the same general pattern of results was qualitatively present even when drastically reducing the size of the dataset to include only 10 patients. We do not believe that the lack of significance here is surprising, given the limited sample. We have adjusted our text to temper our claim and to more conservatively explain the result.

Finally, to test whether our effects were due to intersubject differences, we performed these same analyses using only data from subjects with bilateral hippocampal electrodes (N=10). While the *hemisphere × condition* interaction was no longer significant ($p = 0.1$) in this limited dataset (likely due to the large reduction in sample size), the general pattern of the effect was retained (see Supplementary Figure S5C).

Response to Reviewer 4

- My only remaining concern is how the authors have presented the within-subjects analysis in the manuscript. I feel that the phrase “Despite the large reduction in the size of the dataset, we still observed the same general pattern of effects” is overselling the result a bit, given that some of the effects are not significant anymore (i.e. the interaction). This should be acknowledged and the statement toned down accordingly. Perhaps something along the lines of “while the interaction was no longer significant (quite possibly due to the large reduction in sample size), the general pattern of the effect was retained”.

We agree that we were not as careful with our wording here as we should have been. Following the Reviewer’s suggestion, we have modified the text to more conservatively describe the results (see response to Reviewer 2 above).

REVIEWERS' COMMENTS:

Reviewer #4 (Remarks to the Author):

We are happy with the revisions made. All our points have been addressed, hence we recommend to publish this very interesting paper.

Response to issues raised by referees

No new concerns were raised by the referees. The only comment received was from Reviewer 4, who wrote:

- We are happy with the revisions made. All our points have been addressed, hence we recommend to publish this very interesting paper.

We thank the reviewer for their helpful comments throughout this revision process.